# GraphKeeper: Graph Domain-Incremental Learning via Knowledge Disentanglement and Preservation

**Zihao Guo[1], Qingyun Sun[1]\*, Ziwei Zhang[1], Haonan Yuan[1], Huiping Zhuang[2], Xingcheng Fu[3], Jianxin Li[1]**

[1]SKLCCSE, School of Computer Science and Engineering, Beihang University
[2]Shien-Ming Wu School of Intelligent Engineering, South China University of Technology
[3]Key Lab of Education Blockchain and Intelligent Technology, Guangxi Normal University
`{guozh,sunqy,zwzhang,yuanhn,lijx}@buaa.edu.cn`
`hpzhuang@scut.edu.cn, fuxc@gxnu.edu.cn`

## Abstract

Graph incremental learning (GIL), which continuously updates graph models by sequential knowledge acquisition, has garnered significant interest recently. However, existing GIL approaches focus on task-incremental and class-incremental scenarios within a single domain. Graph domain-incremental learning (Domain-IL), aiming at updating models across multiple graph domains, has become critical with the development of graph foundation models (GFMs), but remains unexplored in the literature. In this paper, we propose **Graph** Domain-Incremental Learning via **K**nowledge Dis**e**ntangl**e**ment and **P**res**e**rvation (**GraphKeeper**), to address catastrophic forgetting in Domain-IL scenario from the perspectives of embedding shifts and decision boundary deviations. Specifically, to prevent embedding shifts and confusion across incremental graph domains, we first propose the domain-specific parameter-efficient fine-tuning together with intra- and inter-domain disentanglement objectives. Consequently, to maintain a stable decision boundary, we introduce deviation-free knowledge preservation to continuously fit incremental domains. Additionally, for graphs with unobservable domains, we perform domain-aware distribution discrimination to obtain precise embeddings. Extensive experiments demonstrate the proposed GraphKeeper achieves state-of-the-art results with 6.5%~16.6% improvement over the runner-up with negligible forgetting. Moreover, we show GraphKeeper can be seamlessly integrated with various representative GFMs, highlighting its broad applicative potential.

## 1 Introduction

Graph incremental learning (GIL)[40, 6, 47] aims to continuously update graph models as new graph data arrives, and has attracted increasing attention. Most existing methods target **task-incremental** (Task-IL) and **class-incremental** (Class-IL) settings (Figure 1), where new tasks or classes emerge within a single domain. However, as newly added graphs often come from different domains, the **domain-incremental** (Domain-IL) setting becomes essential. This is especially relevant with the rise of graph foundation models (GFMs), which require integrating diverse graphs from multiple domains to build a comprehensive and evolving knowledge base. Unfortunately, current GIL methods are designed for single-domain scenarios and struggle with Domain-IL. As illustrated in Figure 2, a representative method SSM [45] performs well under Class-IL but fails in the more challenging Domain-IL case. Despite its importance, effective GIL under Domain-IL remains an open problem.

---

*Corresponding author.

39th Conference on Neural Information Processing Systems (NeurIPS 2025).

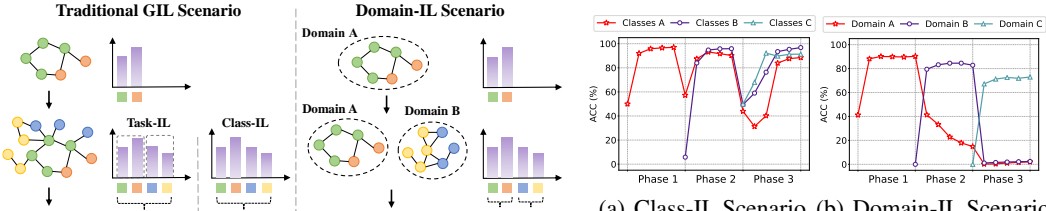

Figure 1: An illustration of traditional GIL (*i.e.*, Task-IL and Class-IL) and Domain-IL scenarios within our scope studied in this paper.

Figure 2: Performance of SSM [45], a representative GIL method, in Class-IL scenario and the more challenging Domain-IL scenario.

(a) Class-IL Scenario (b) Domain-IL Scenario

To tackle this problem, we first analyze the underlying cause of performance degradation in the Domain-IL setting, which is the catastrophic forgetting problem. In typical graph learning models such as GNNs, learning new graphs causes changes in model parameters that manifest in **two aspects**: **(1)** shifts in the embeddings of previously learned graphs, and **(2)** deviations in the decision boundary. These changes together lead to the forgetting of prior knowledge. Existing GIL methods attempt to constrain or adapt to such changes, which is relatively feasible in Task-IL and Class-IL scenarios, as the data still resides within a single domain. In contrast, Domain-IL involves substantial structural and semantic discrepancies across domains, making it much harder for GNNs to retain knowledge across diverse graph distributions. This difficulty is also evidenced by the negative transfer observed in multi-domain graph pre-training [2].

Based on the analysis above, we tackle the GIL challenges in Domain-IL scenario from two sides: **(1) Embedding shifts:** *How to learn stable and disentangled representations for different graph domains in GIL?* The embedding shifts introduced by model parameter changes bring the risk of semantic confusion between incremental graphs. To adapt to new domains, more drastic parameter changes occur, making embedding shifts more uncontrollable. **(2) Decision boundary deviations:** *How to effectively retain knowledge from previously learned graph domains?* The deviation in decision boundary is also a significant cause of catastrophic forgetting [51], especially when the model adapts to new domains. Previous GIL methods, which treat embedding learning and prediction as an integrated process, are difficult to constrain the decision boundary effectively.

In this paper, we propose a novel GIL framework **GraphKeeper** to address the catastrophic forgetting in Domain-IL scenario. Specifically: **(1)** To address the embedding shifts challenge, we propose the domain-specific graph parameter-efficient fine-tuning (PEFT) based on a pre-trained GNN to learn embeddings correspond to graph domains, which ensures parameters for previously learned graph domains remain unaffected when learning new domains. Additionally, we propose intra- and inter-domain disentanglement to avoid confusion in embedding space. **(2)** To address the decision boundary deviation challenge, we separate the decision module from the embedding model and continuously fit incremental domains through the analytical solution of ridge regression, which prevents drastic changes in the decision boundary while effectively retaining knowledge from previously learned graph domains. **(3)** Lastly, we perform domain-aware distribution discrimination on graphs with unobservable domain to facilitate the precise embedding and prediction. **Our contributions are:**

- We propose a novel GIL framework GraphKeeper to tackle the catastrophic forgetting in Domain-IL scenario. To the best of our knowledge, we are the first to explore this challenging problem.
- We design the multi-domain graph disentanglement and deviation-free knowledge preservation mechanism, which enables GraphKeeper to learn stable and disentangled representations between different incremental graph domains and maintains stable decision boundary without deviations.
- GraphKeeper achieves 6.5%∼16.6% improvement over the runner-up with negligible forgetting. Besides, it can be seamlessly integrated with existing representative GFMs.

## 2  Related Work

### 2.1  Graph Incremental Learning

Existing graph incremental learning methods can be categorized into regularization-based, memory replay-based, and parameter isolation-based. Regularization-based methods [16, 31, 4] aim to identify

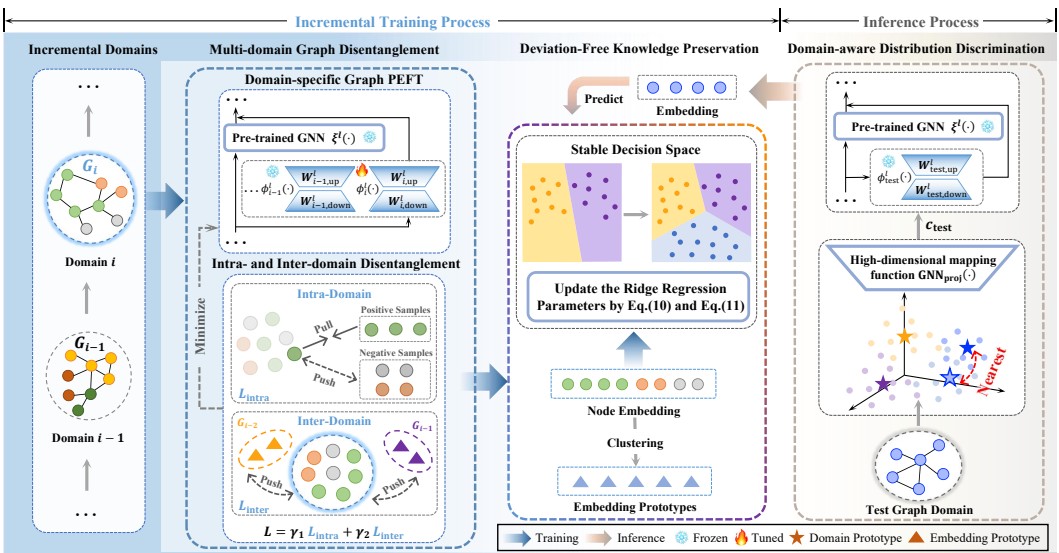

Figure 3: The overall framework of GraphKeeper. **(1)** The Multi-domain Graph Disentanglement isolates parameters of different graph domains through the domain-specific graph PEFT to prevent embedding shifts, and disentangles embeddings both intra- and inter-domain to prevent confusion. **(2)** The Deviation-Free Knowledge Preservation continuously fits incremental graph domains while maintaining a stable decision boundary without deviations. **(3)** The Domain-aware Distribution Discrimination matches graphs with unobservable domain to prototypes of previous domains. Then our method embeds them with corresponding domain-specific PEFT module, and make predictions.

parameters that are important for previous tasks and constrain their changes through penalty terms. Memory replay-based methods [52, 45, 34, 33, 46, 17, 42, 22] are widely recognized for their effectiveness, explicitly preserving representative subsets of graphs from previous tasks and replaying them while learning new tasks. Parameter isolation-based methods [44, 41, 23] adapt to graph data in new tasks through adding trainable parameters. Despite the impressive performance in Task-IL and Class-IL scenarios, these methods do not consider the unique challenges in the Domain-IL scenario.

## 2.2 Multi-domain Graph Learning

**Cross-domain Graph Pre-training.** A promising approach for constructing GFMs is to pre-train on a large corpus of graphs, and transfer the pre-trained knowledge to downstream graphs [48]. Although existing cross-domain graph pre-training methods [49, 39, 37, 50] can effectively adapt to downstream graphs, the model becomes fixed once pre-training is complete. Continuously learning knowledge from new graphs is essential [47], especially when dealing with multiple downstream graph domains, but it faces the problem of catastrophic forgetting. Our method addresses the challenging Domain-IL scenario in GIL and can be integrated with various graph pre-training methods.

**Graph Domain Adaptation.** Graph domain adaptation (GDA) [28] aims to leverage knowledge from the source domain to improve the performance in the target domain. Although multiple domains are considered, GDA and Domain-IL scenario in GIL are fundamentally different [36]. For GDA, the source and target domains are simultaneously accessible in most cases, with the focus primarily on target domain performance. In comparison, for Domain-IL scenario, only the domain of current task is accessible, and the objective is to maintain performance across all previously learned domains.

## 3 Preliminary and Problem Formulation

**Graph Incremental Learning.** Given a task sequence $S = \{G_1, \ldots, G_T\}$, the objective of GIL is to sequentially learn the graphs while ensuring that the model parameters $\boldsymbol{\theta}_t$ after learning the current task maintain satisfactory performance on all previous tasks. The objective can be formulated as:

$$\boldsymbol{\theta}_t^* = \arg\min_{\boldsymbol{\theta}} \frac{1}{t} \sum_{i=1}^{t} \mathcal{L}_i(\boldsymbol{\theta}),$$ (1)

where $\mathcal{L}_i$ is $i$-th task loss. Previous $t-1$ tasks and their data are inaccessible when learning the $t$-th.

**Problem Formulation.** We primarily focus on GIL in the Domain-IL scenario, *i.e.*, graphs $G_t$ in $S$ belong to different domains. This contrasts with Task-IL and Class-IL, where all tasks belong to the same domain. Due to a significant domain gap in both structure and feature semantics, conflicts arise among incremental domains in Domain-IL scenario. Our goal is to learn an encoder $\Phi : \mathcal{V} \mapsto \mathcal{Z}$, where different domains are disentangled in the embedding space $\mathcal{Z}$, representing their respective semantics, and to learn a decoder $\Psi : \mathcal{Z} \mapsto \mathcal{P}$ that accurately maps the disentangled embedding of different graph domains to the label space $\mathcal{P}$, while continuously updating without forgetting.

# 4 Method

In this section, we present GraphKeeper to address the catastrophic forgetting in the Domain-IL scenario. Our key insight is to decompose the cause of catastrophic forgetting into embedding shifts and decision boundary deviations. The framework is shown in Figure 3. In brief, we first propose multi-domain graph disentanglement (Sec. 4.1) to prevent embedding shifts and confusion across incremental graph domains. Then, we introduce deviation-free knowledge preservation (Sec. 4.2) to continuously fit incremental domains while maintaining a stable decision boundary without deviations. Lastly, we design a domain-aware distribution discrimination mechanism (Sec. 4.3) for graphs with unobservable domains to obtain precise embeddings.

## 4.1 Multi-domain Graph Disentanglement

To prevent confusion across incremental graph domains, we introduce multi-domain graph disentanglement, which isolates parameters between different graph domains through the domain-specific graph PEFT to avoid embedding shifts, and disentangles embeddings both intra- and inter-domain.

**Multi-domain Graph Feature Alignment.** Prior to further processing, we first align the feature dimension for graphs from different domains:

$$\tilde{\boldsymbol{F}}_i = \mathrm{Proj}\left(\boldsymbol{F}_i\right), \quad \tilde{\boldsymbol{F}}_i \in \mathbb{R}^{|G_i| \times \overline{d}}, \tag{2}$$

where $\boldsymbol{F}_i$ is the original node features of $G_i$, $|G_i|$ denotes the number of nodes, $\mathrm{Proj}(\cdot)$ represents a projection operation, and $\overline{d}$ represents the unified feature projection dimension. The projection operation $\mathrm{Proj}(\cdot)$ is realized through truncated singular value decomposition (SVD) [29].

In addition to this dimension alignment, the feature semantics of different domains remain fundamentally distinct. On the one hand, drastic changes in model parameters can occur when adapting to new domains, causing the embedding shifts across graph domains. On the other hand, the embeddings of different graph domains may exhibit unintentional overlap that contradicts their semantics, leading to confusion in prediction. Both factors can lead to the catastrophic forgetting in the Domain-IL scenario. To address these issues, we introduce two core modules as follows.

**Domain-specific Graph PEFT.** To prevent potential catastrophic forgetting of previous domains due to drastic parameter changes when adapting to new domains, we aim to isolate parameters of different domains. Inspired by the success of low-rank adaptation (LoRA) [10, 14, 9, 38], we propose graph domain-specific PEFT. Specifically, for graph domain in the task sequence, we equip the pre-trained GNN with a LoRA module. For the $i$-th domain, the $l$-th layer of the model $\psi_i$ is calculated as:

$$\mathbf{h}^l = \boldsymbol{\xi}^l(\mathbf{h}^{l-1}, \boldsymbol{W}_i^l) + \boldsymbol{\phi}_i^l(\mathbf{h}^{l-1}, \boldsymbol{W}_{i,\mathrm{down}}^l \boldsymbol{W}_{i,\mathrm{up}}^l), \tag{3}$$

where $\boldsymbol{\xi}^l(\cdot)$ denotes the output of the frozen GNN, $\boldsymbol{\phi}_i^l(\cdot)$ denotes the LoRA module of the $i$-th graph domain, and $\boldsymbol{W}_{i,\mathrm{down}}^l \in \mathbb{R}^{d^{l-1} \times r}$ and $\boldsymbol{W}_{i,\mathrm{up}}^l \in \mathbb{R}^{r \times d^l}$ are two low-rank learnable parameter matrices, *i.e.*, $r$ is much smaller than the parameter matrix dimension $d^{l-1}$ and $d^l$. When learning a new graph domain, we freeze previous domain-specific LoRA parameters, ensuring that learned graph domains remain stable in the embedding space and do not shift with the adaptation to the new domain.

**Intra- and Inter-domain Disentanglement.** To distinguish semantic differences between different graph domains as well as different classes within the same graph domain, we propose objectives for both intra- and inter-domain disentanglement, aiming to prevent confusion.

**(1) Intra-domain Disentanglement.** Within a single domain, our goal is to enhance the discriminability of node embeddings across different classes. To achieve this, we introduce an intra-domain

disentanglement objective based on contrastive learning. Specifically, we first generate augmented views for a given $G_i$:

$$G_i^{\text{aug}} = \text{Aug}(G_i), \quad \boldsymbol{X}_i = \boldsymbol{\psi}_i(G_i), \quad \boldsymbol{X}_i^{\text{aug}} = \boldsymbol{\psi}_i(G_i^{\text{aug}}), \tag{4}$$

where $\text{Aug}(\cdot)$ denotes feature and structure augmentation, $\boldsymbol{X}_i$ and $\boldsymbol{X}_i^{\text{aug}}$ are learned node embeddings corresponding to the original view and the augmented view, respectively, and $\boldsymbol{\psi}_i$ represents the domain-specific PEFT module in Eq. (3). For each node, we define the positive samples $\boldsymbol{S}^{\text{pos}}$ as the set of nodes belonging to the same class, and the negative samples $\boldsymbol{S}^{\text{neg}}$ as the nodes from different classes. The **intra-domain** disentanglement objective can be formulated as:

$$\mathcal{L}_{\text{intra}} = -\sum_{j=1}^{|G_i|} \log \frac{\sum_{o \in \boldsymbol{S}_j^{\text{pos}}} \exp(\text{sim}(\boldsymbol{x}_j, \boldsymbol{x}_o^{\text{aug}}))}{\sum_{o' \in \boldsymbol{S}_j^{\text{pos}} \cup \boldsymbol{S}_j^{\text{neg}}} \exp(\text{sim}(\boldsymbol{x}_j, \boldsymbol{x}_{o'}^{\text{aug}}))}, \tag{5}$$

where $\text{sim}(\cdot, \cdot)$ denotes cosine similarity. This objective promotes semantic similarity among nodes of the same class while maintaining distinction between nodes of different classes in a single domain.

**(2) Inter-domain Disentanglement.** With the intra-domain disentanglement, the embeddings for node of each class within a domain are compact to ensure clear discriminability between classes. However, in the Domain-IL scenario, due to the feature semantics gap across different graph domains, there exist risks of unintentional overlap in the embedding space across domains. To prevent catastrophic forgetting caused by this confusion between domains, we next introduce the representation scattering based inter-domain disentanglement objective.

Through freezing of domain-specific LoRA parameters, the embeddings of previous domains remain stable. Consequently, the inter-domain disentanglement objective can be defined as ensuring that the current domain is sufficiently distant from previous domains in the embedding space. However, previous graphs are not accessible in GIL. Considering that the embeddings within each domain are compact and unchanged, after learning an incremental domain, we capture representative embedding prototypes through clustering, which represent the distribution of domains in the embedding space. Therefore, the **inter-domain** disentanglement objective can be expressed by pushing the samples of the current domain away from the embedding prototypes of all previous domains:

$$\mathcal{L}_{\text{inter}} = \frac{1}{|G_i|} \sum_{j=1}^{|G_i|} \sum_{k=1}^{|\boldsymbol{P}|} \frac{1}{\|\boldsymbol{x}_j - \boldsymbol{P}_k\|_2^2 + \epsilon}, \tag{6}$$

where $\boldsymbol{P}$ represents the set of embedding prototypes, and $\epsilon$ is a small constant. By minimizing the inter-domain disentanglement objective, the semantics of different domains are clearly separated, with no interference between them.

**Overall Objective.** The overall optimization objective can be formulated as:

$$\mathcal{L} = \gamma_1 \mathcal{L}_{\text{intra}} + \gamma_2 \mathcal{L}_{\text{inter}}, \tag{7}$$

where $\gamma_1$ and $\gamma_2$ are hyperparameters for trading-off. By minimizing Eq. (7), we can obtain highly discriminative node embeddings without confusion for both across different domains and across classes within each domain, thereby facilitating accurate knowledge preservation.

### 4.2 Deviation-Free Knowledge Preservation

Previous GIL methods usually integrate embedding learning and classification through an end-to-end training framework. However, when learning new tasks, the classifier is updated together with the embedding model through back-propagation, leading to the deviation in the decision boundary. Although previous methods have attempted to solve this issue through techniques like memory replay, the limited quantity of memory data is insufficient when faced with the drastic parameter changes required to adapt to new domains.

**Ridge Regression for Stable Classifier Updates.** We aim to solve the decision boundary deviation issue by separating the classifier from the embedding model, thereby maintaining a stable decision boundary that remains unaffected by new incremental domains, which also aligns with the stable embeddings we introduce in the last section. To this end, we introduce a ridge regression-based knowledge preservation mechanism, which does not require gradient updates through back-propagation,

thus avoiding the catastrophic forgetting caused by decision boundary deviations, inspired by [54, 53]. Specifically, for the first incremental domain, denote the stable and disentangled embedding as $\boldsymbol{X}_1$. Then, we fit the class labels $\boldsymbol{Y}_1$ through ridge regression, which is equivalent to:

$$\arg\min_{\boldsymbol{W}_1} \|\boldsymbol{Y}_1 - \boldsymbol{X}_1\boldsymbol{W}_1\|_{\mathrm{F}}^2 + \lambda \|\boldsymbol{W}_1\|_{\mathrm{F}}^2, \tag{8}$$

where $\|\cdot\|_{\mathrm{F}}^2$ indicates the Frobenius norm and $\lambda$ is the regularization coefficient. We can obtain a closed-form solution for the optimal weight as $\boldsymbol{W}_1 = (\boldsymbol{X}_1^\top\boldsymbol{X}_1 + \lambda\boldsymbol{I})^{-1}\boldsymbol{X}_1^\top\boldsymbol{Y}_1$.

Similarly, for the $i$-th incremental domain, the optimal ridge regression parameters $\boldsymbol{W}_i$, which can also retain the knowledge of the previous $i$ domains, can be calculated as:

$$\boldsymbol{W}_i = \left(\boldsymbol{X}_{(1:i)}^\top\boldsymbol{X}_{(1:i)} + \lambda\boldsymbol{I}\right)^{-1}\boldsymbol{X}_{(1:i)}^\top\boldsymbol{Y}_{(1:i)}. \tag{9}$$

**Recursive Update without Historical Data Access.** However, historical domains are not accessible in GIL, *i.e.*, we cannot directly calculate Eq. (9). To solve this, we recursively update the parameter matrix only using data from the current graph domain:

$$\boldsymbol{W}_i = \left[\boldsymbol{W}_{i-1} - \boldsymbol{M}_i\boldsymbol{X}_k^\top\boldsymbol{X}_i\boldsymbol{W}_{i-1} \,\|\, \boldsymbol{M}_i\boldsymbol{X}_i^\top\boldsymbol{Y}_i\right], \tag{10}$$

where $\|$ indicates the concatenation and $\boldsymbol{M}_i$ is an intermediate matrix recursively updated as:

$$\boldsymbol{M}_i = \boldsymbol{M}_{i-1} - \boldsymbol{M}_{i-1}\boldsymbol{X}_i^\top\left(\boldsymbol{I} + \boldsymbol{X}_i\boldsymbol{M}_{i-1}\boldsymbol{X}_i^\top\right)^{-1}\boldsymbol{X}_i\boldsymbol{M}_{i-1}, \tag{11}$$

where $\boldsymbol{M}_1 = (\boldsymbol{X}_1^\top\boldsymbol{X}_1 + \lambda\boldsymbol{I})^{-1}$. The detailed derivations are provided in Appendix A.

Through Eq. (10) and Eq. (11), we can precisely update the model parameters without using historical data while guaranteeing the optimal solution as Eq. (9). For inference, given a graph $G_k$ from the $k$-th domain, we can obtain the predicted node labels as:

$$\boldsymbol{Y}_{\mathrm{predict}} = \mathrm{Softmax}(\boldsymbol{X}_k\boldsymbol{W}), \tag{12}$$

where $\boldsymbol{X}_k$ is calculated by Eq. (3). For graph $G_{\mathrm{test}}$ with unobservable domain, the corresponding domain-specific PEFT module $\phi_{\mathrm{test}}$ in Eq. (3) is determined by the domain-aware distribution discrimination in Sec. 4.3. Since both the embeddings and the decision boundary remain stable, this significantly reduces the risk of catastrophic forgetting caused by drastic parameter changes.

### 4.3 Domain-aware Distribution Discrimination

To obtain precise embeddings, it is necessary to match the graphs from different domains with the corresponding PEFT module, which is easy to achieve for the training process, as the domains of the training graphs are observable. However, for the test graph with an unobservable domain, it is difficult to directly match with the corresponding PEFT module. Therefore, we propose a simple yet effective method for domain-aware distribution discrimination. The core idea is to match test graphs with prototypes of different domains.

Primarily, the features across multiple graph domains are at risk of being excessively similar or even overlapping, as shown in Figure G.3, which may cause the test graph to be matched with the wrong domain prototype. To reduce the risk of prototype confusion, we aim to obtain domain prototypes with sufficient discriminability. Specifically, we first transform the features into a high-dimensional space through a randomly initialized and frozen GNN for random mapping:

$$\hat{\boldsymbol{F}} = \mathrm{GNN}_{\mathrm{proj}}(G, \boldsymbol{W}_{\mathrm{random}}), \tag{13}$$

where $\hat{\boldsymbol{F}}$ denotes the transformed features. Though the $\mathrm{GNN}_{\mathrm{proj}}$ is not trained, this high-dimensional random mapping [30, 19] helps to separate prototypes between different domains, particularly when utilizing the feature distance as the criterion for domain distribution discrimination. The confusion matrix of domain prototypes before and after mapping by Eq. (13) is shown in Figure E.2.

Then, we determine the domain of the test graph by associating it with the nearest domain prototype. Specifically, we calculate the correlation between the test graph and domain prototypes based on the distances, and the domain with the highest correlation is chosen as the domain of the test graph:

$$\boldsymbol{D}_k = \frac{1}{|G_k|}\sum_{i=1}^{|G_k|}\hat{\boldsymbol{F}}_i^k, \quad c_{\mathrm{test}} = \arg\max_k(\exp(-\|\boldsymbol{D}_{\mathrm{test}} - \boldsymbol{D}_k\|_2^2)), \tag{14}$$

where $c_{\text{test}}$ indicates the domain index, $\boldsymbol{D}_{\text{test}}$ and $\boldsymbol{D}_k$ are the prototype of the test and $k$-th incremental graph domain, obtained through average pooling.Our high-dimensional random mapping guarantees a sufficiently distance between the prototypes of different domains, which results in a minimal correlation between the prototype of the test graph and the prototypes of other domains, effectively preventing confusion in domain discrimination.

The overall algorithm and complexity analysis of GraphKeeper are shown in Appendix B.

# 5 Experiments

## 5.1 Experimental setup

**Datasets and Baselines.** To evaluate GraphKeeper[2], we conduct comprehensive experiments on 15 real-world datasets, where detailed descriptions are in Appendix C.1. We choose a variety of baselines, including general IL methods EWC [13], MAS [1], GEM [18], LWF [15] and representative GIL methods TWP [16], ER-GNN [52], SSM [45], DeLoMe [22], PDGNNs [42], TPP [23]. Additionally, we include two self-designed baselines: Fine-Tune, which continuously training without any IL techniques, and Joint, which utilizes all previous graph domains for training.

**Implementation Details.** We implement all methods under the GIL benchmark [43]. All results are averaged over 5 independent runs for a fair comparison. More details are provided in Appendix D.

**Experimental Settings.** To comprehensively evaluate in the Domain-IL scenario, we adopt multiple groups of graph domains. Due to the significant differences between domains, the performance varies to some extent under different incremental orders. Therefore, we set multiple incremental orders for each group and report the average results. Further details are provided in Appendix C.2.

**Evaluation Metrics.** We adopt two commonly used metrics in GIL: Average Accuracy (AA) and Average Forgetting (AF). Higher AA indicates better performance, and higher AF indicates less forgetting, as detailed in Appendix C.3.

## 5.2 Performance on Domain-IL Scenario

**Analysis.** We can observe the following findings from the results of Domain-IL scenario in Table 1:

- While existing GIL methods can handle Task-IL and Class-IL scenarios, they show a significant decline in performance in Domain-IL scenario, which indicates the inability of traditional GIL methods to adapt to the more challenging Domain-IL scenario. In contrast, GraphKeeper achieves state-of-the-art results with 6.5%∼16.6% improvement over the runner-up and with negligible forgetting, which benefits from our disentangled and stable representation for different incremental graph domains through the domain-specific graph PEFT to prevent confusion, and performing high-fidelity knowledge preservation.
- GraphKeeper outperforms the Joint baseline even though it has access to all previous graph domains. The results indicate that a single GNN struggles to effectively integrate knowledge from multiple domains. In contrast, GraphKeeper tackles this issue by isolating parameters of different domains through the domain-specific graph PEFT.
- DeLoMe and PDGNNs exhibit relatively passable performance. These methods utilize SGC [35] and APPNP [8] as their backbones, respectively, which sacrifices plasticity to mitigate forgetting. However, this limits their flexibility and applicability. We also try replacing their backbones as GCN and the results are summarized in Table E.3, showing a significant decline in performance.
- TPP utilizes prompts to adapt to different incremental graphs. However, its capability for modeling incremental graph domains is relatively constrained.

## 5.3 Studies of Integration with GFMs

To further explore the potential of GraphKeeper, we integrate it with two representative GFMs, GCOPE [49] and MDGPT [39], to equip them with the continuous updating capability. Specifically, we first pre-train the GFMs and then incorporate GraphKeeper with them to evaluate the performance in the **few-shot** Domain-IL scenario. The results are summarized in Table 2.

---

[2] https://github.com/RingBDStack/GraphKeeper

Table 1: The average performance (%) across different incremental orders in Domain-IL scenario. AA indicates Average Accuracy and AF indicates Average Forgetting. ± represents the standard deviation. The best results are indicated in **bold** and the runner-ups are underlined.

| Method | Group 1 | | Group 2 | | Group 3 | | Group 4 | | Group 5 | | Group 6 | |
|---|---|---|---|---|---|---|---|---|---|---|---|---|
| | AA ↑ | AF ↑ | AA ↑ | AF ↑ | AA ↑ | AF ↑ | AA ↑ | AF ↑ | AA ↑ | AF ↑ | AA ↑ | AF ↑ |
| Fine-Tune | 23.9±0.1 | -72.7±0.3 | 17.1±0.4 | -74.4±0.7 | 20.9±0.1 | -81.3±0.2 | 21.6±0.6 | -77.8±1.1 | 19.7±0.1 | -77.8±0.3 | 18.6±0.6 | -77.9±0.6 |
| Joint | 66.6±0.8 | - | 67.6±0.4 | - | 78.0±0.3 | - | 75.7±0.8 | - | 74.5±0.3 | - | 71.5±0.5 | - |
| EWC | 23.3±0.1 | -72.0±0.3 | 17.3±0.4 | -74.8±1.0 | 20.8±0.3 | -80.9±0.2 | 22.4±0.6 | -76.1±0.8 | 20.6±0.2 | -77.9±0.6 | 18.0±0.3 | -76.6±0.9 |
| MAS | 24.0±2.4 | -70.5±4.3 | 16.8±5.1 | -69.8±5.0 | 18.9±2.5 | -77.4±3.0 | 21.1±5.0 | -75.5±7.7 | 18.9±3.4 | -74.0±3.2 | 19.8±2.3 | -71.5±1.1 |
| GEM | 23.3±3.2 | -71.7±5.6 | 20.1±3.4 | -69.2±2.8 | 21.7±2.3 | -80.6±2.1 | 21.1±4.2 | -77.4±4.3 | 20.0±2.5 | -77.4±2.1 | 19.4±1.8 | -76.1±0.7 |
| LWF | 23.7±2.9 | -72.5±1.3 | 17.0±0.7 | -73.8±0.7 | 20.2±0.2 | -80.7±0.2 | 20.1±0.2 | -78.6±0.6 | 19.8±0.2 | -77.9±0.4 | 18.8±0.3 | -76.0±0.7 |
| TWP | 23.5±0.1 | -71.0±0.5 | 16.0±3.3 | -71.0±1.6 | 20.0±2.2 | -79.0±0.7 | 20.2±2.0 | -79.0±1.0 | 18.8±3.4 | -74.4±1.2 | 18.2±2.1 | -75.2±1.0 |
| ER-GNN | 23.3±1.4 | -64.8±3.8 | 22.7±2.2 | -66.3±2.9 | 28.7±3.0 | -66.3±3.7 | 23.9±1.7 | -72.9±2.8 | 24.8±2.7 | -68.8±3.9 | 24.7±4.7 | -67.9±5.8 |
| SSM | 24.1±7.8 | -36.4±13.1 | 22.1±3.6 | -35.7±5.8 | 15.6±3.3 | -31.0±4.1 | 25.3±2.6 | -45.7±3.8 | 25.7±2.8 | -39.3±3.3 | 16.1±5.7 | -33.9±4.8 |
| TPP | 52.6±1.8 | 0.0±0.0 | 49.7±1.5 | 0.0±0.0 | 57.1±2.5 | 0.0±0.0 | 53.0±2.9 | 0.0±0.0 | 56.7±1.0 | 0.0±0.0 | 48.3±2.4 | 0.0±0.0 |
| DeLoMe | 49.3±0.1 | -7.7±0.1 | 58.2±5.1 | -9.6±8.5 | 70.2±4.1 | -1.0±0.1 | 73.4±4.3 | -1.1±0.2 | 63.2±5.7 | -3.3±7.1 | 64.2±2.2 | -4.8±5.9 |
| PDGNNs | 52.4±0.5 | -15.8±0.6 | 53.5±0.3 | -7.4±0.3 | 65.5±0.6 | -11.4±0.7 | 65.5±0.5 | -7.2±0.8 | 64.3±0.3 | -8.2±0.3 | 60.2±0.2 | -7.2±0.4 |
| **GraphKeeper** | **69.2±0.3** | -0.4±0.2 | **73.1±1.1** | -2.8±0.7 | **80.6±0.4** | 0.0±0.1 | **79.9±0.5** | 1.0±0.5 | **75.5±0.9** | 0.1±0.5 | **77.5±0.7** | -0.1±0.5 |

Table 2: The results of integrating GraphKeeper with existing GFMs in the **few-shot** Domain-IL senario. AA indicates Average Accuracy and AF indicates Average Forgetting. ± represents the standard deviation. Better results are indicated in **bold**.

| Method | Group 1 | | Group 2 | | Group 3 | | Group 4 | | Group 5 | | Group 6 | |
|---|---|---|---|---|---|---|---|---|---|---|---|---|
| | AA ↑ | AF ↑ | AA ↑ | AF ↑ | AA ↑ | AF ↑ | AA ↑ | AF ↑ | AA ↑ | AF ↑ | AA ↑ | AF ↑ |
| GCOPE | 20.6±0.9 | -53.5±2.1 | 10.5±0.7 | -36.7±1.0 | 13.2±1.2 | -47.7±2.9 | 12.6±1.1 | -51.0±1.8 | 13.6±1.4 | -41.7±2.9 | 12.6±0.9 | -43.7±1.1 |
| GCOPE+Ours | **56.8±1.9** | 0.2±0.3 | **36.6±1.4** | 0.3±0.3 | **47.4±4.1** | -1.6±2.8 | **51.6±1.6** | -0.4±0.6 | **44.3±4.0** | 0.8±0.5 | **44.9±0.9** | -0.8±0.8 |
| MDGPT | 19.9±1.3 | -59.4±2.1 | 10.7±0.5 | -42.7±2.2 | 12.1±1.5 | -50.4±2.1 | 12.8±1.4 | -52.9±3.1 | 12.8±1.3 | -47.6±5.9 | 12.3±0.5 | -48.8±3.3 |
| MDGPT+Ours | **59.7±1.2** | -1.5±0.7 | **32.7±1.0** | -0.4±0.7 | **49.5±2.1** | -2.8±1.1 | **62.7±1.7** | -1.4±1.2 | **50.9±1.6** | -0.7±0.4 | **43.9±1.1** | -0.2±1.0 |

**Analysis.** We observe that the original GFMs exhibit low AA and high AF. The high AF indicates that they perform well when trained separately on each graph domain, demonstrating strong few-shot capability. However, they lack the continuous updating capability with severe catastrophic forgetting issues, resulting in low AA. After integrating the GraphKeeper, they show significantly higher AA with negligible forgetting. This demonstrates that GraphKeeper can be seamlessly integrated into existing GFMs, combining their few-shot strength with the continuous updating capability to construct more powerful GFMs. Additionally, GraphKeeper does not require memory replay, which avoids the potential memory explosion problem, particularly on a large corpus of graphs.

## 5.4 Ablation Study

**Ablation Study.** We analyze three variants of Graph-Keeper. The results are summarized in Figure 4 and Figure E.1.

- **GraphKeeper (w/o DT)**: We replace the disentanglement objectives in Eq. (7) with the vanilla contrastive objective that ignores domain-specific semantics.
- **GraphKeeper (w/o PEFT)**: We remove the domain-specific graph PEFT module and perform continuous training on the pre-trained GNN model.
- **GraphKeeper (w/o KP)**: We remove the deviation-free knowledge preservation module and utilize an MLP as the classifier for end-to-end training.

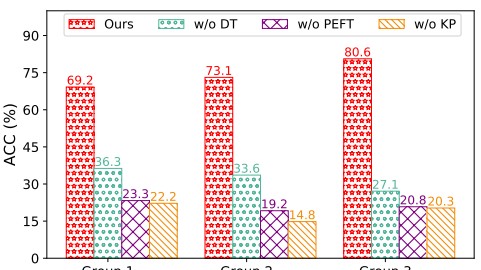

Figure 4: The ablation study results.

**Analysis.** Overall, we observe that removing any key component of GraphKeeper results in a noticeable drop in performance, highlighting the importance of each module. **(1)** For GraphKeeper (w/o

DT), removing both intra-domain and inter-domain disentanglement objectives significantly weakens the model's ability to distinguish between classes and domains. This results in entangled embeddings that cause confusion across domains and hinder generalization. **(2)** For GraphKeeper (w/o PEFT), excluding parameter-efficient fine-tuning removes the isolation of domain-specific parameters. As a result, adaptation to new domains induces large parameter updates that affect previously learned domains, leading to embedding shifts and inconsistency with the preserved knowledge. **(3)** For GraphKeeper (w/o KP), the absence of knowledge preservation allows the classifier parameters to drift throughout training. This drift causes deviations in the decision boundary, disrupting the alignment between the learned embeddings and their corresponding labels. These results confirm that both embedding stability and decision boundary consistency are essential to mitigate catastrophic forgetting in the Domain-IL setting, and that GraphKeeper achieves this through the synergistic effect of disentanglement, parameter isolation, and knowledge preservation.

## 5.5 Visualisation of Disentangled Embeddings

Embedding confusion is an important cause of catastrophic forgetting in Domain-IL scenario. To provide a more intuitive comparison of the advantages of GraphKeeper, we visualize the embeddings of different incremental graph domains with t-SNE [32] in Figure G.3 and Figure G.4.

**Analysis.** As shown in Figure G.3, the embeddings produced by GraphKeeper are highly compact with clear boundaries between domains, thanks to the introduction of multi-domain graph disentanglement, which effectively reduces interference across incremental domains. In contrast, PDGNNs and DeLoMe exhibit noticeable domain overlap. Although they replay previous data, they still struggle to avoid conflicts between different domains. To further investigate the evolutions, we also visualize previously learned domains at different learning stages (Figure G.4). It can be observed that the embeddings of PDGNNs and DeLoMe become increasingly entangled over time, while those of GraphKeeper maintain high separability throughout, further corroborating our analysis.

## 5.6 Hyperparameters Analysis

We analyze the impact of hyperparameters in Figure 5, where $r$ denotes the rank of the parameter matrix in domain-specific graph PEFT module, and $\gamma_1/\gamma_2$ plays the trade-off role between intra- and inter-domain disentanglement objectives.

**Analysis. (1)** For hyperparameter $r$, a small value limits the number of tunable parameters, making it difficult for the model to adapt effectively to new graph domains. As $r$ increases, the performance improves and achieves strong results while still using far fewer parameters than full-model tuning. **(2)** For hyperparameters $\gamma_1$ and $\gamma_2$, when the ratio $\gamma_1/\gamma_2$ decreases, meaning the inter-domain disentanglement objective becomes dominant, the model shows relatively poor performance due to insufficient class discrimination within each domain. On the other hand, when the intra-domain objective dominates, the performance also degrades because the lack of inter-domain disentanglement causes partial confusion between domains. A balanced trade-off between $\gamma_1$ and $\gamma_2$ yields the best performance.

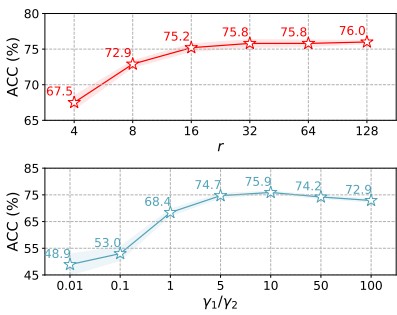

Figure 5: Hyperparameter analysis.

## 6 Conclusion

In this paper, we propose a novel GIL framework, GraphKeeper, which effectively addresses catastrophic forgetting in the Domain-IL setting by mitigating embedding shifts and decision boundary deviations. The multi-domain graph disentanglement module learns stable and disentangled representations, thereby reducing interference among domains. In parallel, the deviation-free knowledge preservation mechanism maintains a consistent decision boundary across tasks. Moreover, the domain-aware distribution discrimination enables precise embedding of graphs with unknown domain labels. Extensive experiments validate the effectiveness of GraphKeeper and highlight its strong potential for seamless integration into graph foundation models.

## Acknowledgements

The corresponding author is Qingyun Sun. Authors of this paper are supported by the National Natural Science Foundation of China through grants No.62302023 and No.62225202, the Fundamental Research Funds for the Central Universities JK2024-07, and the Guangzhou Basic and Applied Basic Research Foundation 2024A04J3681. We owe sincere thanks to all authors for their valuable efforts and contributions.

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

# A  Derivation of Eq. (10) and Eq. (11)

We first restate Eq. (10) and Eq. (11) for reference.

---

**Eq. (10):**
$$W_i = \left[W_{i-1} - M_i X_k^\top X_i W_{i-1} \,\|\, M_i X_i^\top Y_i\right],$$
where $\|$ indicates the concatenation and $M_i$ is an intermediate matrix.
**Eq. (11):**
$$M_i = M_{i-1} - M_{i-1} X_i^\top \left(I + X_i M_{i-1} X_i^\top\right)^{-1} X_i M_{i-1},$$
where $M_1 = (X_1^\top X_1 + \lambda I)^{-1}$.

---

*Proof.* We begin by defining the closed-form solution of ridge regression at step $k$ as:

$$W_k = M_k X_{(1:k)}^\top Y_{(1:k)}. \tag{A.1}$$

where $M_k = \left(X_{(1:k)}^\top X_{(1:k)} + \lambda I\right)^{-1}$. This can be recursively expressed using the matrix inversion lemma as:

$$M_k = \left(M_{k-1}^{-1} + X_k^\top X_k\right)^{-1}. \tag{A.2}$$

Applying the Woodbury matrix identity, we derive the recursive update of $M_k$:

$$M_k = M_{k-1} - M_{k-1} X_k^\top \left(I + X_k M_{k-1} X_k^\top\right)^{-1} X_k M_{k-1}, \tag{A.3}$$

which corresponds to Eq. (11). We then turn to the update of $W_k$. By definition:

$$
\begin{aligned}
W_k &= M_k X_{(1:k)}^\top Y_{(1:k)} \\
&= M_k \left[X_{(1:k-1)}^\top Y_{(1:k-1)} \,\|\, X_k^\top Y_k\right] \\
&= \left[M_k X_{(1:k-1)}^\top Y_{(1:k-1)} \,\|\, M_k X_k^\top Y_k\right].
\end{aligned} \tag{A.4}
$$

We now focus on computing the first term. Substituting Eq. (A.3) into this term yields:

$$
\begin{aligned}
M_k X_{(1:k-1)}^\top Y_{(1:k-1)} &= M_{k-1} X_{(1:k-1)}^\top Y_{(1:k-1)} \\
&\quad - M_{k-1} X_k^\top \left(I + X_k M_{k-1} X_k^\top\right)^{-1} X_k M_{k-1} X_{(1:k-1)}^\top Y_{(1:k-1)} \\
&= W_{k-1} - M_{k-1} X_k^\top \left(I + X_k M_{k-1} X_k^\top\right)^{-1} X_k W_{k-1}.
\end{aligned} \tag{A.5}
$$

To simplify the matrix expression, we define:

$$K_k = \left(I + X_k M_{k-1} X_k^\top\right)^{-1}. \tag{A.6}$$

From the identity $K_k \left(I + X_k M_{k-1} X_k^\top\right) = I$, we can derive:

$$K_k = I - K_k X_k M_{k-1} X_k^\top. \tag{A.7}$$

Using this, we rewrite the matrix product as:

$$
\begin{aligned}
M_{k-1} X_k^\top \left(I + X_k M_{k-1} X_k^\top\right)^{-1} &= M_{k-1} X_k^\top K_k \\
&= M_k X_k^\top.
\end{aligned} \tag{A.8}
$$

Substituting Eq. (A.8) into Eq. (A.5) gives:

$$M_k X_{(1:k-1)}^\top Y_{(1:k-1)} = W_{k-1} - M_k X_k^\top X_k W_{k-1}. \tag{A.9}$$

Finally, substituting Eq. (A.9) back into Eq. (A.4), we obtain:

$$W_k = \left[W_{k-1} - M_k X_k^\top X_k W_{k-1} \,\|\, M_k X_k^\top Y_k\right], \tag{A.10}$$

where $M_k$ is recursively updated via Eq. (A.3).

This completes the derivation of Eq. (10) and Eq. (11). $\qquad\square$

# B   Algorithms and Comlexity Analysis

## B.1   Algorithms

The overall training and inference process of GraphKeeper are given in Algorithm 1 and Algorithm 2.

---
**Algorithm 1:** The overall training process of GraphKeeper

---

**Input:** Graph domain sequence $S = \{G_1, \ldots, G_T\}$; Pre-trained GNN $\boldsymbol{\xi}$; Number of training epochs $E$.

**Output:** Ridge regression parameters $\boldsymbol{W}$; Domain-specific LoRA parameters $\{\boldsymbol{\phi}_1, \ldots, \boldsymbol{\phi}_T\}$; Domain prototypes $\{\boldsymbol{D}_1, \ldots, \boldsymbol{D}_T\}$; Random projection function $\text{GNN}_{\text{proj}}$.

**1** Randomly initialize and frozen the parameters of $\text{GNN}_{\text{proj}}$;
**2** Frozen the parameters of $\boldsymbol{\xi}$;
**3** Initialize the set $\boldsymbol{P}$ of the embedding prototypes;
**4** **for** $t = 1, \ldots, T$ **do**
    // Multi-domain Graph Disentanglement
**5**     Align the feature dimensions of $G_t$ by Eq. (2);
**6**     Initialize the LoRA parameters $\boldsymbol{\phi}_t$ for $G_t$;
**7**     **for** $epoch = 1, \ldots, E$ **do**
**8**         Train the LoRA parameters $\boldsymbol{\phi}_t$ by Eq. (7);
**9**     **end**
**10**     Calculate the embedding $\boldsymbol{X}_t$ by Eq. (3);
**11**     Add new embedding prototypes into set $\boldsymbol{P}$ by clustering;
    // Deviation-Free Knowledge Preservation
**12**     Update the ridge regression parameters $\boldsymbol{W}$ by Eq. (10) and (11);
**13**     Calculate the domain prototype $\boldsymbol{D}_t$;
**14** **end**

---

---
**Algorithm 2:** The inference process of GraphKeeper

---

**Input:** Test graph $G_{\text{test}}$; Ridge regression parameters $\boldsymbol{W}$; Pre-trained GNN $\boldsymbol{\xi}$; Domain-specific LoRA parameters $\{\boldsymbol{\phi}_1, \ldots, \boldsymbol{\phi}_T\}$; Domain prototypes $\{\boldsymbol{D}_1, \ldots, \boldsymbol{D}_T\}$; Random projection function $\text{GNN}_{\text{proj}}$.

**Output:** Predicted result of the test graph $G_{\text{test}}$.

    // Domain-aware Distribution Discrimination
**1** Calculate the test prototype $\boldsymbol{D}_{\text{test}}$;
**2** Discriminate the domain of $G_{\text{test}}$ by Eq. (14);
    // Inference
**3** Calculate the embedding $\boldsymbol{X}_{\text{test}}$ with $\boldsymbol{\xi}$ and $\boldsymbol{\phi}_{\text{test}}$ by Eq. (3);
**4** Get the predicted result by Eq. (12);

---

## B.2   Comlexity Analysis

For brevity, $\bar{d}$ donates the unified feature dimension, $|G_i|$ donates the number of nodes in the $i$-th graph domain, and $d$ donates the model's hidden layer dimension. For a single graph domain, with the model containing 2 layers, the complexity of the forward propagation of the frozen pre-trained GNN is $\mathcal{O}(|G|\bar{d}d + |G|d^2)$, and the complexity of LoRA forward and backward propagation is $\mathcal{O}(2(|G|(\bar{d}r + rd) + 2|G|(dr)) = \mathcal{O}(2|G|r(\bar{d} + 3d))$, where $r$ is the rank of LoRA module. The complexity of intra-domain disentanglement loss calculated by Eq. (5) is $\mathcal{O}(|G|^2 d)$, and the complexity of inter-domain disentanglement loss calculated by Eq. (6) is $\mathcal{O}(|G||P|d)$, where $|P|$ donates the number of embedding prototypes. Given the number of the training epochs $E$, the combined complexity of these components is $\mathcal{O}(E(|G|(d(\bar{d} + d + 6r + |G| + |P|) + 2r\bar{d})))$. For Eq. (10) and Eq. (11), the primary computational cost arises from matrix inversion, with a complexity of $\mathcal{O}(|G|^3)$, but this is performed only once. The overall computational complexity is $\mathcal{O}(\sum_{i=1}^{T} E(|G_i|(d(\bar{d} + d + 6r + |G_i| + |P|) + 2r\bar{d})) + |G_i|^3)$. As the feature dimension $\bar{d}$, the rank $r$, hidden layer dimension $d$, the number of embedding prototypes $|P|$, and the training epochs $E$ are relatively small compared to

the number of nodes, the overall computational complexity can then be approximately reduced to $\mathcal{O}(\sum_{i=1}^{T}(|G_i|^2 + |G_i|^3)) = \mathcal{O}(\sum_{i=1}^{T}|G_i|^3)$

# C  Experiment Details

## C.1  Datasets

We conduct experiments on 15 real-world datasets, including academic networks (Cora [26], Citeseer [12], PubMed [21], CoauthorCS [27], and DBLP [7]), co-purchase networks (Photo [27] and Computer [27]), web networks (WikiCS [20], Facebook [24], Chameleon [24], and Squirrel [24]), social networks (GitHub [24], LastFMAsia [25], and DeezerEurope [25]), and airline networks (Airport [3]). Statistics of datasets are concluded in Table C.1. All the datasets are consented to by the authors for academic usage. All the datasets do not contain personally identifable information or offensive content.

Table C.1: Statistics of datasets.

| Dataset | # Nodes | # Edges | # Features | # Classes | # Homophily |
|---|---|---|---|---|---|
| Cora | 2,708 | 5,429 | 1,433 | 7 | 0.81 |
| Citeseer | 3,327 | 4,732 | 3,703 | 6 | 0.74 |
| PubMed | 19,717 | 44,338 | 500 | 3 | 0.80 |
| CoauthorCS | 18,333 | 163,788 | 6,805 | 15 | 0.81 |
| DBLP | 17,716 | 105,734 | 1,639 | 4 | 0.83 |
| Photo | 7,650 | 119,081 | 745 | 8 | 0.83 |
| Computer | 13,752 | 245,778 | 767 | 10 | 0.78 |
| WikiCS | 11,701 | 431,726 | 300 | 10 | 0.65 |
| Facebook | 22,470 | 342,004 | 128 | 4 | 0.89 |
| Chameleon | 2,277 | 36,101 | 2,325 | 5 | 0.24 |
| Squirrel | 5,201 | 217,073 | 2,089 | 5 | 0.22 |
| GitHub | 37,700 | 578,006 | 128 | 2 | 0.85 |
| LastFMAsia | 7,624 | 55,612 | 128 | 18 | 0.87 |
| DeezerEurope | 28,281 | 185,504 | 128 | 2 | 0.53 |
| Airport | 3,188 | 18,631 | 4 | 4 | 0.72 |

## C.2  Experiment Setting

### C.2.1  Domain-IL Scenario

To comprehensively evaluate the effectiveness of methods in Domain-IL scenario, we obtained 6 increment groups of graph domain from 15 datasets, as shown in Table C.2. Group 1 to Group 3 consist of graph domains from the same type (*e.g.*, all are social networks in a group), while Group 4 to Group 6 include graph domains from mixed types. For the group containing graph domains A, B, C, and D, we evaluate in incremental orders of A → B → C → D, B → C → D → A, C → D → A → B, and D → A → B → C, respectively. For each graph domain, we set the unified dimension of the features to 512 and split the training set, validation set, and test set in proportions of 60%, 20%, and 20%. For the few-shot setting, we sample 10 labeled nodes from all classes within each graph domain for training.

## C.3  Evaluation Metrics

The accuracy matrix is denoted as $M$, where $M_{i,j}$ represents the accuracy on domain $j$ after learning domain $i$. We adopt two commonly used metrics: Average Accuracy (AA) and Average Forgetting (AF). The AA is computed as $\frac{1}{T}\sum_{j=1}^{T}M_{T,j}$, which represents the average performance across all domains after learning all domains. On the other hand, AF quantifies forgetting across domains through $\frac{1}{T-1}\sum_{j=1}^{T-1}(M_{T,j} - M_{j,j})$, which represents the difference between the final performance on each domain and the performance on the domain when it was first learned. Higher AA indicate better performance and higher AF indicates less forgetting.

Table C.2: Statistics of incremental groups of graph domain.

| Group | Graph Domains |
|---|---|
| Group 1 | GitHub, LastFMAsia, DeezerEurope |
| Group 2 | WikiCS, Facebook, Chameleon, Squirrel |
| Group 3 | Citeseer, Pubmed, CoauthorCS, DBLP |
| Group 4 | Pubmed, Photo, WikiCS, Airport |
| Group 5 | CoauthorCS, Computer, Chameleon, DeezerEurope |
| Group 6 | Cora, Facebook, LastFMAsia, Squirrel |
| Group 7 | GitHub, LastFMAsia, DeezerEurope, WikiCS, Facebook, Chameleon, Squirrel, Citeseer, Pubmed, CoauthorCS, DBLP |
| Group 8 | Pubmed, Photo, WikiCS, Airport, CoauthorCS, Computer, DeezerEurope, Cora, Facebook, LastFMAsia, Squirrel |

## C.4 Running Environment

We conduct the experiments with:

- Operating System: Ubuntu 20.04 LTS.
- CPU: Intel(R) Xeon(R) Platinum 8358 CPU@2.60GHz with 1TB DDR4 of Memory.
- GPU: NVIDIA Tesla V100 with 32GB of Memory.
- Software: CUDA 11.7, Python 3.8.0, Pytorch 1.7.1, DGL 0.6.1.

# D  Implementation Details

## D.1  Implementation Details of GraphKeeper

We set the number of model layers to 2 for all methods, with the learning rate set to 5e-2, the weight decay coefficient set to 5e-4, and 200 training epochs per incremental graph domain. The parameters of models are optimized by Adam [11]. For GraphKeeper, the pre-trained GNN model is trained through link prediction on the first incremental domain (except for Sec. 5.3), the feature and structure augmentation are achieved by randomly masking a few features and dropping a few edges, and the embedding prototype sampling is implemented through DBSCAN [5]. The hyperparameter $r$ is chosen from $\{4, 8, 16, 32, 64, 128\}$, $\gamma_1$ is chosen from $\{0.01, 0.1, 1.0, 5.0, 10.0\}$, $\gamma_2$ is chosen from $\{0.001, 0.01, 0.10, 0.50, 1.0\}$.

## D.2  Implementation Details of Baselines

We implement all methods under the GIL benchmark [43].

- **EWC** [13], **MAS** [1], **GEM** [18], **LWF** [15], **TWP** [16], **ER-GNN** [52]: `https://github.com/QueuQ/CGLB` [CC BY 4.0 License].
- **SSM** [45]: `https://github.com/QueuQ/SSM` [CC BY 4.0 License].
- **TPP** [23]: `https://github.com/mala-lab/TPP` [MIT License].
- **DeLoMe** [22]: `https://github.com/mala-lab/DeLoMe` [with license unspecified].
- **PDGNNs** [42]: `https://github.com/imZHANGxikun/PDGNNs` [CC BY 4.0 License].

# E  Additional Experiment Results

## E.1  Comparison of Methods with Unified GCN Backbone

Considering that some baselines adopt simplified GNN such as SGC [35] and APPNP [8] as their backbone, they achieve stability at the cost of sacrificing model plasticity. However, constraining the backbone of the model imposes limitations on practical applications, especially when the model architecture cannot be replaced. Thus, we unity the backbone of related methods to GCN for a fair

Table E.3: Comparison of performance with the unified GCN backbone.

| Method | Group 1 | | Group 2 | | Group 3 | | Group 4 | | Group 5 | | Group 6 | |
|---|---|---|---|---|---|---|---|---|---|---|---|---|
| | AA ↑ | AF ↑ | AA ↑ | AF ↑ | AA ↑ | AF ↑ | AA ↑ | AF ↑ | AA ↑ | AF ↑ | AA ↑ | AF ↑ |
| DeLoMe | 49.2±2.1 | -20.9±3.3 | 32.2±1.4 | -51.8±1.7 | 56.9±0.3 | -19.8±0.3 | 40.9±1.3 | -44.8±1.6 | 49.1±1.3 | -31.7±2.1 | 45.4±1.5 | -35.3±2.4 |
| PDGNNs | 48.0±2.4 | -33.4±3.8 | 42.3±2.5 | -39.1±2.9 | 53.4±0.7 | -35.5±0.9 | 40.9±1.7 | -49.4±2.4 | 49.6±1.8 | -36.1±2.5 | 45.1±1.5 | -38.1±2.3 |
| **GraphKeeper** | **69.2±0.3** | -0.4±0.2 | **73.1±1.1** | -2.8±0.7 | **80.6±0.4** | 0.0±0.1 | **79.9±0.5** | 1.0±0.5 | **75.5±0.9** | 0.1±0.5 | **77.5±0.7** | -0.1±0.5 |

Table E.4: Performance comparison on longer Domain-IL sequences.

| Method | Group 7 (11 Domains) | | Group 8 (12 Domains) | |
|---|---|---|---|---|
| | AA ↑ | AF ↑ | AA ↑ | AF ↑ |
| TPP | 50.2±0.9 | 0.0±0.0 | 52.8±2.5 | 0.0±0.0 |
| DeLoMe | > 1 day | | > 1 day | |
| PDGNNs | 37.7±0.1 | -13.1±0.4 | 45.0±0.2 | -13.6±0.3 |
| **GraphKeeper** | **73.0±1.1** | -0.7±0.5 | **76.5±0.9** | -0.8±0.2 |

comparison. The results are summarized in Table E.3. It show that after unifying the backbone, the performance of the baselines declines to a certain extent, significantly reducing their usability. This also highlights the advantage of our approach, which imposes no restrictions on model architecture, achieving a dual benefit of plasticity and stability.

## E.2 Longer Incremental Learning Sequence

We conduct evaluation on two longer incremental sequences, Group 7 and Group 8, containing 11 and 12 graph domains, respectively, which are more challenging. As shown in Table E.4, the advanced baselines suffer significant performance drops, while GraphKeeper maintains strong performance and outperforms the runner-up by 22.8%~23.7%. This demonstrates the scalability of GraphKeeper and its robustness to larger data scales.

## E.3 Ablation study

Due to space limitations, the ablation study from Group 4 to Group 6 are presented in Figure E.1, showing that removing any component of Graph-Keeper leads to a degradation of the performance, especially for GraphKeeper (w/o PEFT) and Graph-Keeper (w/o KP) variants. Specifically, the full model consistently outperforms all ablated variants across the three groups. Notably, removing PEFT or KP causes the most significant drops (*e.g.*, from 79.9% to 21.3%/19.7% in Group 4), highlighting their crucial roles in preserving prior knowledge and supporting cross-domain alignment. The performance of w/o DT is also inferior, confirming the importance of domain transfer modules. These results verify the necessity of each component for robust generalization.

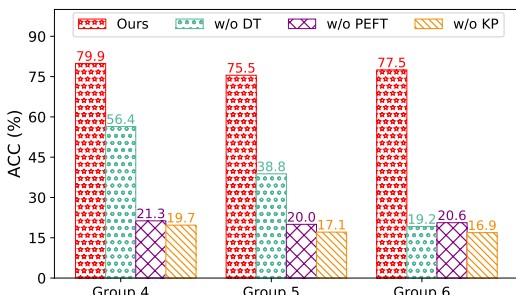

Figure E.1: Ablation study.

## E.4 Domain Discrimination

Through random high-dimensional mapping, we achieve linear separability between domain features that may potentially overlap. As shown in Figure E.2, all domains exhibit a significant reduction in pairwise confusion after mapping. This provides a stability guarantee for our subsequent domain prototype matching.

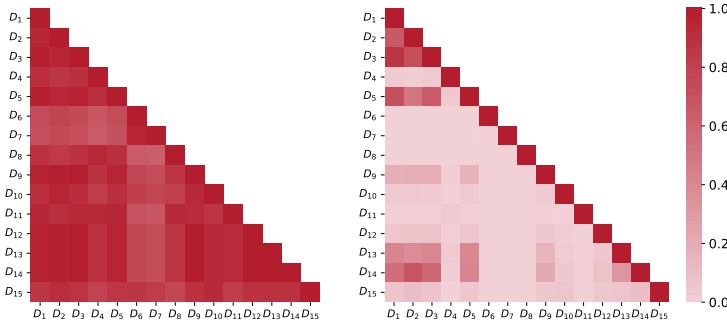

Figure E.2: Confusion matrix of domain prototypes. Deeper color indicates higher proximity. *Left:* Original. *Right:* Ours.

# F    Further Discussion on Related Work

In Sec. 4.3, we introduce the domain-aware distribution discrimination to match graphs with unobservable domains to previous domain prototypes. Recent work [23] discusses a similar issue, they propose applying Laplace smoothing on original features and matching the test graph with the class prototypes through cosine similarity. Unfortunately, as shown in Figure G.3, the features across multiple domains are at risk of being excessively similar or even overlapping, which may cause the test graph to be matched with the wrong domain prototype. We guarantees a sufficiently distance between the prototypes of different domains through high-dimensional random mapping, use distance metrics in high-dimensional space as the criteria for matching, effectively preventing confusion in domain discrimination.

# G    Limitations

The limitations of GraphKeeper include the following aspects. First, when evaluating the performance of GraphKeeper in Domain-IL scenario, we followe the traditional setting, where the classes in the new incremental domain are assumed to be entirely novel. However, more general and practical cases, such as partially overlapping classes across domains, have not been thoroughly explored. Additionally, although GraphKeeper can be seamlessly integrated into existing GFMs, enhancing their continual updating capabilities based on their few-shot learning strengths, it does not fully leverage the zero-shot learning potential of GFMs. This limitation is tied to the scenario design, as it does not account for previously unseen domains, an aspect that has been overlooked by all existing methods. In summary, future work will extend the GraphKeeper's applicability, and investigate more general scenario in GIL.

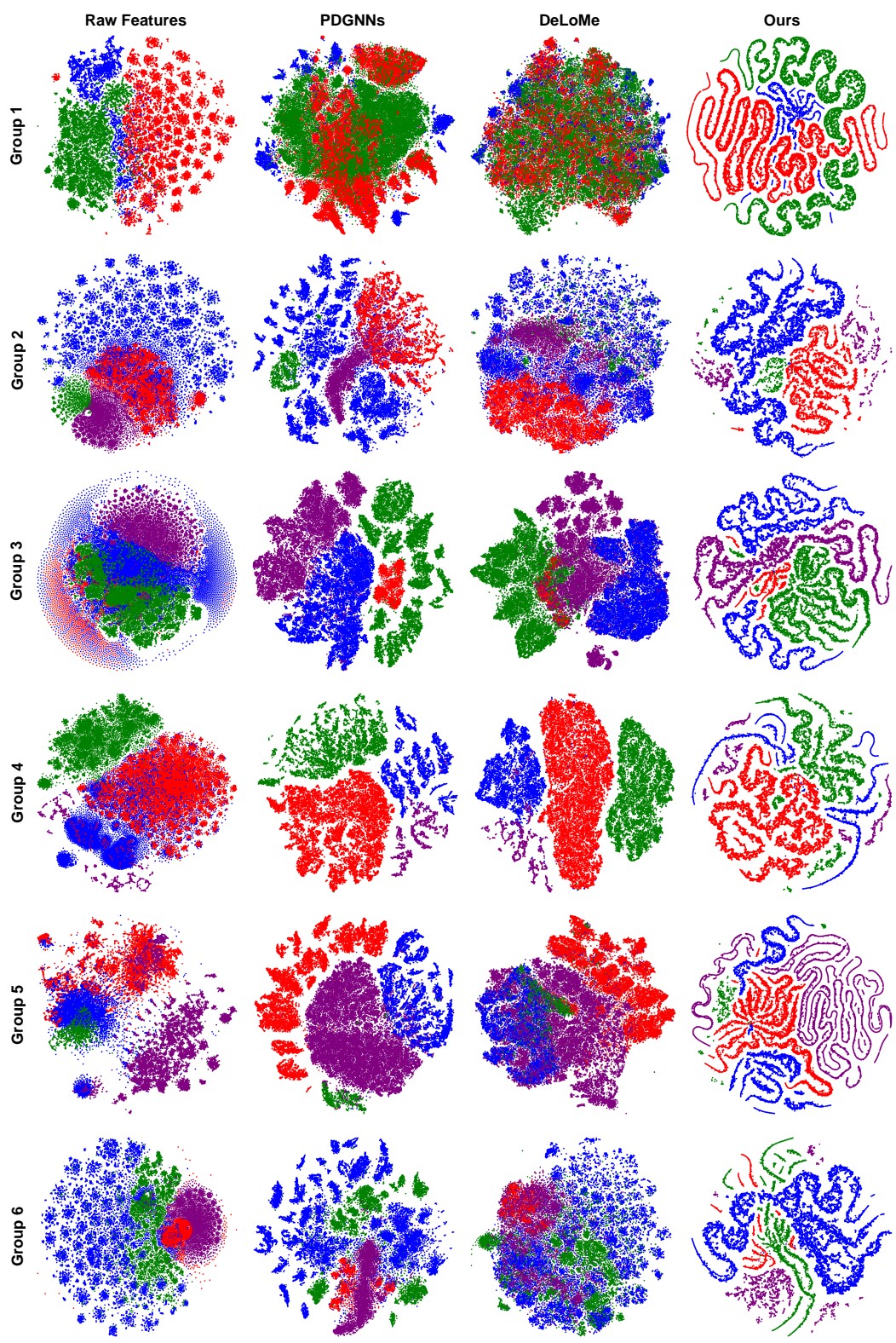

Figure G.3: Visualization of the node embeddings of different graph domains after incrementally learning all graph domains. Each color corresponds to a graph domain. The embeddings produced by GraphKeeper are highly compact with clear boundaries between different graph domains.

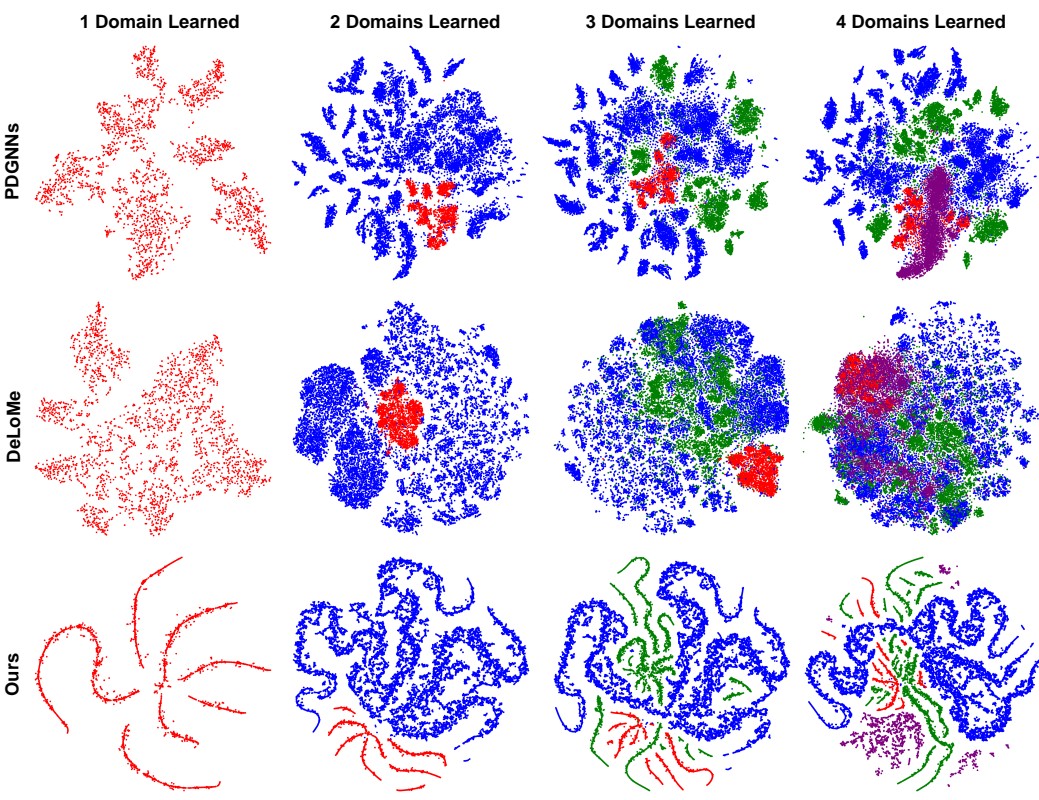

Figure G.4: Visualization of the node embeddings of previously learned domains at different stages of the learning process on Group 6.

