# OpenReview forum: "GraphKeeper: Graph Domain-Incremental Learning via Knowledge Disentanglement and Preservation"
_NeurIPS.cc/2025/Conference — NeurIPS 2025 poster_

### Official Review · Reviewer_CzLz · 2025-06-18

**Clarity:** 2
**Significance:** 3
**Originality:** 2
**Rating:** 4
**Confidence:** 2

**Summary:**

This paper designs GraphKeeper for graph domain incremental learning. It tackles the challenge of catastrophic forgetting by mitigating embedding shifts and decision boundary deviations. GraphKeeper employs parameter-efficient fine-tuning and intra/inter-domain objectives, along with deviation-free knowledge preservation via ridge regression. Furthermore, it incorporates domain-aware distribution discrimination using high-dimensional random mapping. Experiments across 15 real-world datasets demonstrate its performance.

**Questions:**

Please see the above weaknesses.

**Ethical Concerns:**

["NO or VERY MINOR ethics concerns only"]

**Final Justification:**

The analysis of the graph domain incremental learning task is praiseworthy. The author's response resolved my concerns about the method's innovation. So I raised the score from 3 to 4 accordingly.

**Limitations:**

Yes

**Quality:**

3

**Strengths And Weaknesses:**

Strengths:
1. The analysis of the graph domain incremental learning task is praiseworthy, with the challenges well defined.
2. The method's components appear effective, and the experiments are fairly comprehensive. The testing on the GFM model and the ablation experiments are conducted.

Weaknesses:
1. The author proposed a series of components to address the graph domain incremental learning task. But the components are not specially designed for graph data. LoRA-based learning, contrastive learning, and ridge regression can also be adapted to other data types. I worry about the innovativeness of this method for graph domain incremental learning tasks.
2. In formula 7, why are there two loss weights instead of one?
3. In formula 2, the prediction is made using a weight matrix and input. But why is there only a weight matrix and no bias matrix?
4. In formula 13, how can random weight matrices compute stable and accurate node representations to get domain prototypes?
5. In Figure 1, what's the accuracy of the proposed method in predicting domain prototypes? If using domain-specific parameters, whether the forgetting rate is zero like the TPP [1] method?

[1] Replay-and-forget-free graph class-incremental learning: A task profiling and prompting approach. NeurIPS 2024.

---

> ### Author Rebuttal · Authors · 2025-07-28
>
> We appreciate your comments. We provide responses below to address your concerns.
>
> ---
>
> > **Q1: More Discussion on Method Design and Novelty (Weakness 1)**
> >
>
> **A1:** We appreciate the reviewer’s concern. Firstly, our method is designed around the significant structural and semantic differences across graph domains, as well as the limitations of vanilla GNN architectures in capturing multi-domain knowledge, which are the core challenges for graph data and models in Domain-IL scenario. A more detailed analysis of topology will be explored in future work. Additionally, the main contribution of this paper focuses on addressing the problems in the challenging scenario we proposed. While some techniques employed are not originally proposed in this paper, we have specifically adapted and refined them based on the characteristics of the scenario, making them suitable for graph data and models, and effectively addressing the challenges in Domain-IL scenario.
>
> ---
>
> > **Q2: The explanation of Formulas 7 and Formulas 12 (Weakness 2 & 3)**
> >
>
> **A2:** We are grateful for the reviewer's concern to this detail.
>
> - **Regarding Formula 7:** it is mathematically equivalent to the single-weight form $\mathcal{L} = \gamma \mathcal{L}\_{\text{intra}} + (1-\gamma) \mathcal{L}\_{\text{inter}}$, as both serve to balance the contributions of the intra-domain and inter-domain disentanglement objectives.
> - **Regarding Formula 12:** the bias term is omitted for mathematical and operational consistency. The weight matrix $W$ is derived via ridge regression in Eq. (9) is recursively updated in Eq. (10) and Eq. (11) to optimally fit all incremental graph domains without requiring historical data. Crucially, adding a bias term would disrupt the closed-form solution of ridge regression and the recursive update derivation in Appendix A, which is one of the key points to ensure that the weight matrix $W$ can recursively fit the new domain without forgetting the previous domains.
>
> ---
>
> > **Q3: The explanation of Formulas 13 and Domain-aware Distribution Discrimination (Weakness 4)**
> >
>
> **A3:** We appreciate the reviewer's insightful comment. The random weight matrices in Eq. (13) compute stable domain prototypes through high-dimensional random projection, rather than pursuing precise node representations. The frozen, randomly initialized $GNN_{proj}$ non-linearly projects raw features into a high-dimensional space. This scattering effect amplifies differences between domains while suppressing potiential semantic overlaps, as shown in Figure 4 of the paper. Crucially, the fixed random weights ensure deterministic mappings across all domains, guaranteeing stability. Prototypes derived from averaged embeddings in Eq. (14) thus reliably discriminate domains based on relative distances.
>
> In addition, we also provide theoretical theorems and supplementary experiments below.
>
> **Theorem 1. Linear Separability via Random High-Dimensional Mapping:** Let $D_i$ and $D_j$ be feature distributions of two graph domains with overlap $\kappa \in [0,1]$. For a fixed random mapping $\phi:\mathbb{R}^{d} \rightarrow \mathbb{R}^{m}$, where $m \gg d$, the mapped features satisfy:
>
> $$
> \mathbb{P}\left[\left\| \phi(\boldsymbol{x}_i \sim \mathcal{D}_i) - \phi(\boldsymbol{x}_j \sim \mathcal{D}_j) \right\|_2 \geq \Delta \right] \geq 1 - \delta
> $$
>
> where $\Delta >0$ is a separation margin, and $\delta$ decays exponentially with $m$. Even when $D_i$ and $D_j$ overlap to a large extent, random projection amplifies distributional differences in high-dimensional space, ensuring linear separability with high probability if $\kappa \neq 1$. This stems from the Johnson-Lindenstrauss Lemma and geometric properties of random projections.
>
> **Theorem 2. Stability of Prototype Matching:** Let $\boldsymbol{x}\_{\text{test}} \sim \mathcal{D}\_{k}$ be the test node features from domain $k$, and $\boldsymbol{p}\_{k} = \mathbb{E}\_{\boldsymbol{x}\_{\text{train}} \sim \mathcal{D}\_{k}}[\phi(\boldsymbol{x}\_{\text{train}})]$ be the true domain prototype. Given a fixed random mapping $\phi$ and identical train and test distributions:
>
> $$
> \left\| \boldsymbol{p}_{\text{test}} - \boldsymbol{p}_k \right\|_2 \leq \epsilon_1
> $$
>
> For mismatched domain prototypes $\boldsymbol{p}_{j}$, where $j \neq k$:
>
> $$
> \left\| \boldsymbol{p}_{\text{test}} - \boldsymbol{p}_j \right\|_2 \geq \Delta - \epsilon_2
> $$
>
> where $\Delta$ is the separation margin from Theorem 1. Since the test graph's nodes share the same distribution as the training nodes of its source domain, test graphs converge to their true domain prototype $\boldsymbol{p}\_k$ (i.e. $\epsilon_1$ tends towards zero). Meanwhile, even when the feature distributions across domains exhibit substantial overlap but are not identical (i.e. $\kappa \neq 1$), coupling this with a sufficiently high mapping dimension ensures that $\Delta - \epsilon_2$ tends to be sufficiently large. This guarantees that the lower bound of $\left\| \boldsymbol{p}\_{\text{test}} - \boldsymbol{p}\_j \right\|\_2$ is always greater than the upper bound of $\left\| \boldsymbol{p}\_{\text{test}} - \boldsymbol{p}\_k \right\|\_2$.
>
> ---
>
> > **Q4: The accuracy in predicting domain prototypes and the performance of forgetting rate (Weakness 5)**
> >
>
> **A4:** We appreciate the reviewer's insightful comment.
>
> **(1) Regarding the accuracy of domain prototype prediction:**
> - As illustrated in Figure 4 of the paper, our high-dimensional random mapping significantly enhances the discriminability between domain prototypes. This ensures precise matching between any test graph’s prototype and its belonging domain prototype, effectively eliminating inter-domain semantic confusion and guaranteeing reliable domain discrimination during testing.
>
> - To further verify this, we also provide experimental validation of robustness under extreme conditions (i.e. under different domains with high distributional overlap). Specifically, we randomly sampled graphs from four mutually similar distributions as four graph domains, where the pairwise domains feature distribution overlap exceeded 80%. For each domain, we partitioned training and test nodes to derive domain prototypes and test graph prototypes. The confusion matrix below shows the probability of our method correctly predicting the source domain of each test graph.
>
> |  | $\boldsymbol{p}_{1}$ | $\boldsymbol{p}_{2}$ | $\boldsymbol{p}_{3}$ | $\boldsymbol{p}_{4}$ |
> | --- | --- | --- | --- | --- |
> | $\boldsymbol{p}_{1}^{\text{test}}$ | **38.16%** | 22.77% | 18.91% | 20.14% |
> | $\boldsymbol{p}_{2}^{\text{test}}$ | 21.02% | **38.94%** | 21.30% | 18.72% |
> | $\boldsymbol{p}_{3}^{\text{test}}$ | 19.57% | 23.72% | **39.79%** | 16.91% |
> | $\boldsymbol{p}_{4}^{\text{test}}$ | 23.49% | 20.47% | 16.49% | **39.52%** |
>
> - These results demonstrate that even when confronted with highly similar cross-domain distributions, our method ensures that the matching probability between test graph prototypes and their source domain prototypes is significantly higher than that with other domains, thereby enabling accurate and robust domain-specific parameter assignment.
>
> **(2) Regarding the forgetting rate:** GraphKeeper exhibits near-zero forgetting and even demonstrates positive forward transfer on some graph domains. This fundamentally differs from TPP, while TPP achieves zero forgetting by completely isolating prompt parameters and classifiers across incremental tasks under accuracy task-ID prediction, it severely restricts cross-task knowledge transfer. In contrast, GraphKeeper maps embeddings from different domains into a unified semantic space to preserve domain-specific semantics. When new domains are introduced, the boundary adaptively adjusts when recursively updating the ridge regression classifier, triggering implicit positive transfer. It is worth noting that the current framework demonstrates promise for positive transfer among incremental domains, but not yet consummates. We are actively exploring to achieve that and addressing challenges such as domain-invariant knowledge extraction and negative transfer mitigation, which constitutes future work beyond the scope of this work.

---

> > ### Comment · Reviewer_CzLz · 2025-08-07
> >
> > Thanks for the clarification from the authors. I tend to maintain my score. I am still concerned about the innovativeness of the method. In the weakness 1, the authors did not provide more explanations about the special design of graph data. Regarding the other questions, I have received reasonable responses.

---

> > > ### Author Response · Authors · 2025-08-07
> > > **Further response on innovation**
> > >
> > > We appreciate your feedback and recognition of the responses of other issues.
> > >
> > > Regarding the concern about innovativeness, **we note that other reviewers did not raise similar issues. Their positive feedback suggests that the novelty of our method is well-recognized in the context of graph-specific challenges in Domain-IL.**
> > >
> > > While techniques such as LoRA, contrastive learning, and ridge regression are general-purpose, their adaptation to graph data introduces unique challenges, particularly in the setting of Domain-IL, where we explore their effective applications for the first time. To further clarify, we detail these challenges and our contributions as follows:
> > >
> > > - **LoRA for Graph Domain-IL:** Traditional LoRA assumes global low-rank updates for structured dense data (e.g., text), but graph exhibit irregular structural patterns, especially across different domains. Directly applying global low-rank updates risks disrupting domain-specific topologies, leading to uncontrolled parameter drift. To address this, we localize low-rank updates to domain-specific graphs, ensuring awareness of critical topological features within each domain. Additionally, we freeze LoRA parameters from previous domains to prevent inter-domain interference, a departure from the global updates typical of generic LoRA.
> > > - **Contrastive Learning for Graph Disentanglement:** Traditional graph contrastive learning is typically self-supervised, aiming to learn discriminative and generalizable embedding representations. In our Domain-IL scenario, we repurpose this idea to disentangle semantics across different graph domains as well as between classes within the same domain, thereby avoiding knowledge confusion. This fundamentally alters our definitions of positive/negative samples and objectives compared to traditional contrastive learning. Specifically, we employ contrastive learning principles to disentangle knowledge both inter- and intra-domain, ensuring compact and well-separated embeddings for each domain and class, which represent distinct semantics rather than generating unique embeddings for nodes.
> > > - **Recursive Ridge Regression for Incremental Stable Decision Boundary:** Traditional ridge regression maps raw features to class labels under the assumption of i.i.d. samples, ignoring topology-dependent node correlations. We leverage the advantage of ridge regression in optimizing closed-form solutions while integrating it with GNNs' ability to capture topological relationships, thereby establishing accurate mappings from GNN embeddings to class labels. Moreover, traditional ridge regression performs one-time optimal fitting, which is ill-suited for incremental scenarios. Through theoretical derivation and proof, we extend it to an equivalent multi-step fitting process, enabling stable and precise incremental updates collaborate with other components in our method.
> > >
> > > To better address the your concern, we kindly ask for more detailed feedback to clarify specific confusion regarding the innovativeness.
> > > ﻿
> > >
> > > We hope this response underscores our method’s novelty in tackling graph Domain-IL and look forward to further discussion.

---

> > > > ### Comment · Reviewer_CzLz · 2025-08-08
> > > >
> > > > The author's response resolved my concerns about the method's innovation. So I raised the score from 3 to 4 accordingly.

---

> > > > > ### Author Response · Authors · 2025-08-08
> > > > >
> > > > > We sincerely appreciate your valuable feedback and support throughout the review process!

---

> ### Comment · Area_Chair_K6DS · 2025-08-07
>
> Please leave your feedback for the authors' rebuttal to avoid being flagged as an irresponsible reviewer.

---

### Official Review · Reviewer_kEra · 2025-06-25

**Clarity:** 2
**Significance:** 2
**Originality:** 3
**Rating:** 4
**Confidence:** 3

**Summary:**

This paper pioneers graph domain-incremental learning (Domain-IL), where models continuously learn from diverse graph domains unlike traditional task/class-incremental scenarios. The authors trace catastrophic forgetting to embedding shifts and decision boundary deviations when acquiring new domain knowledge. GraphKeeper addresses these issues through three components: domain-specific PEFT with intra/inter-domain disentanglement prevents embedding confusion; ridge regression-based knowledge preservation maintains stable decision boundaries without historical data; high-dimensional random mapping matches unknown test domains to correct modules.

**Questions:**

See above weaknesses

**Ethical Concerns:**

["NO or VERY MINOR ethics concerns only"]

**Final Justification:**

My final justification is recommendation **4** (Borderline Accept) and my confidence is 3
The paper's core technical strengths and rigorous methodology compensate for the relatively limited scope of its evaluation. Although the reviewers raised concerns about the theoretical justification and experimental procedures, the authors' rebuttals and subsequent discussions with the reviewers clarified that the narrow but conclusive evaluation effectively validated the core arguments. While some uncertainty exists regarding the understanding of some technical details, the fundamental soundness of the study is evident. Therefore, the technical contributions sufficiently outweigh the acknowledged limitations to warrant only a grudging recommendation. The authors are advised to cautiously expand the theoretical comparison in future work.

**Limitations:**

Yes

**Quality:**

2

**Strengths And Weaknesses:**

# **Strengths**

**S1: Domain-Specific Parameter Isolation**

Domain-specific graph PEFT using LoRA modules where each incremental domain gets its own low-rank adaptation parameters while freezing previous domains' parameters, preventing cross-domain parameter interference.

**S2: Dual-Level Disentanglement Mechanism**

Novel intra-domain and inter-domain disentanglement objectives that simultaneously enhance class discriminability within each domain through contrastive learning and ensure domain separation by pushing current domain embeddings away from previous domain prototypes.

**S3: Gradient-Free Decision Boundary Preservation**

The framework separates classifier from embedding model and employs ridge regression with recursive matrix updates (Equations 10-11) to maintain stable decision boundaries without gradient-based updates, eliminating decision boundary drift.

# **Weaknesses**

**W1: Missing Theoretical Convergence Analysis**

The paper lacks theoretical analysis for recursive ridge regression (Equations 10-11). Authors should prove convergence conditions, demonstrate equivalence with batch ridge regression, and provide theoretical justification for how LoRA parameter isolation ensures previous domain embedding stability.

**W2: Insufficient Analysis of Domain Discrimination Failures**

Section 4.3's domain-aware distribution discrimination mechanism lacks examination of failure scenarios. Authors should conduct additional experiments addressing: (1) strategies for handling test graphs whose domain features differ significantly from all trained domains; (2) conditions where high-dimensional random mapping fails to separate domain prototypes effectively; (3) quantifying how domain discrimination errors impact overall performance and whether confidence thresholds or rejection mechanisms are necessary for uncertain domain assignments.

---

> ### Author Rebuttal · Authors · 2025-07-28
>
> We are grateful for your positive feedback and detailed comments. We provide responses below to address your concerns.
>
> ---
>
> > **Q1: Supplementary analysis of recursive ridge regression and embedding stability (Weekness 1)**
> >
>
> **A1:** We appreciate the reviewer's insightful comment.
>
> **(1) Analysis of recursive ridge regression:** The detailed derivation proving the equivalence between incremental updated ridge regression (Eq. (10) and Eq. (11)) and global ridge regression (Eq. (9)) has been provided in Appendix A. To illustrate, we list the main derivation steps here:
>
> *Proof.*  We begin by defining the closed-form solution of ridge regression at step $k$ as:
> $$
> \boldsymbol{W}\_{k} =  \boldsymbol{M}\_{k} \boldsymbol{X}\_{(1:k)}^{\top} \boldsymbol{Y}\_{(1:k)}.
> $$
> where $\boldsymbol{M}\_{k}=\big(\boldsymbol{X}\_{(1:k)}^{\top} \boldsymbol{X}\_{(1:k)}+\lambda \boldsymbol{I}\big)^{-1}$. This can be recursively expressed using the matrix inversion lemma as:
> $$
> \boldsymbol{M}\_{k}=\big(\boldsymbol{M}\_{k-1}^{-1} + \boldsymbol{X}\_{k}^{\top} \boldsymbol{X}\_{k}\big)^{-1}.
> $$
> Applying the Woodbury matrix identity, we derive the recursive update of $\boldsymbol{M}\_k$:
> $$
> \boldsymbol{M}\_k = \boldsymbol{M}\_{k-1}
>         - \boldsymbol{M}\_{k-1} \boldsymbol{X}\_k^{\top} \big(\boldsymbol{I} + \boldsymbol{X}\_k \boldsymbol{M}\_{k-1} \boldsymbol{X}\_k^{\top} \big)^{-1} \boldsymbol{X}\_k \boldsymbol{M}\_{k-1}.
> $$
> We then turn to the update of $\boldsymbol{W}\_k$. By definition:
> $$
> \boldsymbol{W}\_{k} = \big[\boldsymbol{M}\_k \boldsymbol{X}\_{(1:k-1)}^{\top} \boldsymbol{Y}\_{(1:k-1)}  ||  \boldsymbol{M}\_k \boldsymbol{X}\_{k}^{\top} \boldsymbol{Y}\_{k}\big],
> $$
> where the first term can be simplified through a series of derivations:
> $$
> \boldsymbol{M}\_k \boldsymbol{X}\_{(1:k-1)}^{\top} \boldsymbol{Y}\_{(1:k-1)} = \boldsymbol{W}\_{k-1} - \boldsymbol{M}\_k \boldsymbol{X}\_k^{\top} \boldsymbol{X}\_k \boldsymbol{W}\_{k-1}.
> $$
> Substituting it back into $\boldsymbol{W}\_{k}$, we obtain:
> $$
> \boldsymbol{W}\_{k} = \big[ \boldsymbol{W}\_{k-1} - \boldsymbol{M}\_k \boldsymbol{X}\_k^{\top} \boldsymbol{X}\_k \boldsymbol{W}\_{k-1} || \boldsymbol{M}\_k \boldsymbol{X}\_{k}^{\top} \boldsymbol{Y}\_{k} \big].
> $$
>
> This proves the equivalence of Eq. (10) and Eq. (11) with global ridge regression in Eq. (9). The complete derivation process can be found in Appendix A.
>
> **(2) Analysis of embedding stability:** Each incremental domain uses domain-specific PEFT modules fine-tuned on a pre-trained GNN backbone. After training, these parameters are frozen and stored. During inference, domain-aware distribution discrimination selects the correct frozen PEFT parameters to combine with the backbone. Since both the pre-trained backbone and domain-specific PEFT parameters remain fixed after training, embeddings for previous domains are stable as long as the domain-aware distribution discriminator assigns the correct PEFT parameters, ensuring stability under the identically distributed data in the same domain. Further explanation can be obtained in Theorem 3 of the response to reviewer #HUJf.
>
> ---
>
> > **Q2: Supplementary analysis of Domain Discrimination (Weekness 2)**
> >
>
> **A2:** We thank the reviewer for raising this insightful concern. Below, we provide a systematic response through theoretical theorems and supplementary experiments. The supplementary analyses will be incorporated into the revised version.
>
> **Theorem 1. Linear Separability via Random High-Dimensional Mapping:** Let $D_i$ and $D_j$ be feature distributions of two graph domains with overlap $\kappa \in [0,1]$. For a fixed random mapping $\phi:\mathbb{R}^{d} \rightarrow \mathbb{R}^{m}$, where $m \gg d$, the mapped features satisfy:
>
> $$
> \mathbb{P}\left[\left\| \phi(\boldsymbol{x}_i \sim \mathcal{D}_i) - \phi(\boldsymbol{x}_j \sim \mathcal{D}_j) \right\|_2 \geq \Delta \right] \geq 1 - \delta
> $$
>
> where $\Delta >0$ is a separation margin, and $\delta$ decays exponentially with $m$. Even when $D_i$ and $D_j$ overlap to a large extent, random projection amplifies distributional differences in high-dimensional space, ensuring linear separability with high probability if $\kappa \neq 1$. This stems from the Johnson-Lindenstrauss Lemma and geometric properties of random projections.
>
> **Theorem 2. Stability of Prototype Matching:** Let $\boldsymbol{x}\_{\text{test}} \sim \mathcal{D}\_{k}$ be the test node features from domain $k$, and $\boldsymbol{p}\_{k} = \mathbb{E}\_{\boldsymbol{x}\_{\text{train}} \sim \mathcal{D}\_{k}}[\phi(\boldsymbol{x}\_{\text{train}})]$ be the true domain prototype. Given a fixed random mapping $\phi$ and identical train and test distributions:
>
> $$
> \left\| \boldsymbol{p}_{\text{test}} - \boldsymbol{p}_k \right\|_2 \leq \epsilon_1
> $$
>
> For mismatched domain prototypes $\boldsymbol{p}_{j}$, where $j \neq k$:
>
> $$
> \left\| \boldsymbol{p}_{\text{test}} - \boldsymbol{p}_j \right\|_2 \geq \Delta - \epsilon_2
> $$
>
> where $\Delta$ is the separation margin from Theorem 1. Since the test graph's nodes share the same distribution as the training nodes of its source domain, test graphs converge to their true domain prototype $\boldsymbol{p}\_k$ (i.e. $\epsilon_1$ tends towards zero). Meanwhile, even when the feature distributions across domains exhibit substantial overlap but are not identical (i.e. $\kappa \neq 1$), coupling this with a sufficiently high mapping dimension ensures that $\Delta - \epsilon_2$ tends to be sufficiently large. This guarantees that the lower bound of $\left\| \boldsymbol{p}\_{\text{test}} - \boldsymbol{p}\_j \right\|\_2$ is always greater than the upper bound of $\left\| \boldsymbol{p}\_{\text{test}} - \boldsymbol{p}\_k \right\|\_2$.
>
> **Supplementary experiments.** We conducted robustness evaluations under different domains with high distributional overlap. Specifically, we randomly sampled graphs from four mutually similar distributions as four graph domains, where the pairwise domains feature distribution overlap exceeded 80%. For each domain, we partitioned training and test nodes to derive domain prototypes and test graph prototypes. The confusion matrix below shows the probability of our method correctly predicting the source domain of each test graph.
>
> ||$\boldsymbol{p}_{1}$|$\boldsymbol{p}_{2}$|$\boldsymbol{p}_{3}$|$\boldsymbol{p}_{4}$|
> |---|---|---|---|---|
> |$\boldsymbol{p}_{1}^{\text{test}}$|**38.16%**|22.77%|18.91%|20.14%|
> |$\boldsymbol{p}_{2}^{\text{test}}$|21.02%|**38.94%**|21.30%|18.72%|
> |$\boldsymbol{p}_{3}^{\text{test}}$|19.57%|23.72%|**39.79%**|16.91%|
> |$\boldsymbol{p}_{4}^{\text{test}}$|23.49%|20.47%|16.49%|**39.52%**|
>
> These results demonstrate that even when confronted with highly similar cross-domain distributions, our method ensures that the matching probability between test graph prototypes and their source domain prototypes is significantly higher than that with other domains, thereby enabling accurate and robust domain-specific parameter assignment.
>
> ---

---

> > ### Comment · Reviewer_kEra · 2025-08-06
> >
> > Thank you for your clarification. For my questions, I have received sufficient answers. Respectfully, I would like to keep my score as is.

---

### Official Review · Reviewer_4hfB · 2025-07-02

**Clarity:** 3
**Significance:** 3
**Originality:** 4
**Rating:** 5
**Confidence:** 4

**Summary:**

The paper introduces GraphKeeper, a framework for Graph Domain-Incremental Learning that prevents catastrophic forgetting by stabilizing embeddings and decision boundaries across graph domains. It combines domain-specific fine-tuning, disentanglement objectives, and knowledge preservation. Experiments show it outperforms existing methods, offering improved accuracy and minimal forgetting for multi-domain graph learning.

**Questions:**

1. How does GraphKeeper perform when dealing with graph data from domains that have very little overlap in their feature semantics? Are there any specific domains where the framework struggles?

2. In the experiments, you compare GraphKeeper with methods like TPP and DeLoMe, which also address catastrophic forgetting. Could you discuss in more detail how GraphKeeper is fundamentally different in terms of model architecture or training procedure?

3. How does the method scale to a large number of domains, and can the PEFT modules be shared or compressed across similar domains?

4. What's the limitation of existing related works? Please declare the majoir difference between GraphKeeper and these.

5. For other questions, please refer to Weakness.

**Ethical Concerns:**

["NO or VERY MINOR ethics concerns only"]

**Final Justification:**

Most of the concerns had been addressed

**Quality:**

4

**Strengths And Weaknesses:**

**Strengths:**
1. GraphKeeper fills a gap in graph incremental learning by addressing the challenge of catastrophic forgetting in domain-incremental learning, which is an area that has been less explored.
2. The combination of domain-specific PEFT, disentanglement objectives (intra- and inter-domain), and deviation-free knowledge preservation offers a comprehensive framewrk to mitigate forgetting and improve model stability across domains.
3. The framework's ability to integrate with existing GFMs enhances its potential for use in real-world applications involving dynamic, multi-domain graph data.

**Weaknesses:**
1. The paper doesn't provide an ablation study for the domain-aware distribution discrimination mechanism (Section 4.3), which could offer deeper insights into its effectiveness.
2. The paper lacks comparison with the CaT [1], which would be a relevant baseline in the context of graph continual learning.
3. The proposed inter-domain loss pushes samples away from previous domain prototypes using inverse distance. This is a heuristic formulation, lacking theoretical grounding on why this regularization ensures optimal knowledge retention.
4. Missing related references, e.g., [2]

[1] Cat: Balanced continual graph learning with graph condensation. ICDM 2023.

[2] Dink-net: Neural clustering on large graphs

---

> ### Author Rebuttal · Authors · 2025-07-28
>
> We appreciate your positive feedback and insightful comments. We provide responses below to address your concerns.
>
> ---
>
> > **Q1: The ablation study for Domain-aware Distribution Discrimination and comparison with CaT (Weakness 1 & 2)**
> >
>
> **A1:** We sincerely thank the reviewer for this valuable suggestion. To validate the effectiveness of domain-aware distribution discrimination, we remove it to obtain GraphKeeper(w/o Discrim) variant. And we further incorporate CaT into the comparison. The supplementary results will be incorporated into the revised version.
>
> | Method | Group 1 |  | Group 2 |  | Group 3 |  | Group 4 |  | Group 5 |  | Group 6 |  |
> | --- | --- | --- | --- | --- | --- | --- | --- | --- | --- | --- | --- | --- |
> |  | **AA ↑** | **AF ↑** | **AA ↑** | **AF ↑** | **AA ↑** | **AF ↑** | **AA ↑** | **AF ↑** | **AA ↑** | **AF ↑** | **AA ↑** | **AF ↑** |
> | CaT | 26.7±3.3 | -39.1±4.8 | 43.3±1.1 | -7.3±3.3 | 40.6±1.6 | -9.8±3.5 | 45.6±1.9 | -31.8±3.7 | 39.1±3.1 | -11.5±4.1 | 47.9±4.9 | -21.7±3.3 |
> | **GraphKeeper** | **69.2±0.3** | -0.4±0.2 | **73.1±1.1** | -2.8±0.7 | **80.6±0.4** | 0.0±0.1 | **79.9±0.5** | 1.0±0.5 | **75.5±0.9** | 0.1±0.5 | **77.5±0.7** | -0.1±0.5 |
> | w/o Discrim | 39.8±1.9 | -31.4±2.3 | 37.1±1.5 | -30.7±2.3 | 58.5±4.8 | -27.5±7.2 | 30.4±4.3 | -38.0±1.3 | 35.6±7.1 | -24.8±2.7 | 38.7±1.6 | -28.6±3.4 |
>
> As shown in the Table:
>
> - The performance of GraphKeeper(w/o Discrim) degrades significantly due to the lack of prior knowledge for associating the graph with corresponding PEFT module, resulting in less precise embeddings.
> - GraphKeeper still achieves the best performance with incorporating CaT into the comparison, as CaT did not consider Domain-IL's core cross-domain challenges.
>
> ---
>
> > **Q2: More detailed explanation about disentanglement mechanism (Weakness 3)**
> >
>
> **A2:**  We appreciate the reviewer's insightful comment. The inter-domain disentanglement is not a standalone heuristic but a synergistic component of GraphKeeper’s unified framework. Intra- and inter-domain disentanglement collocates nodes of distinct classes into unique semantic location, forming compact intra-class clusters with clear boundaries. The domain-specific PEFT then freezes these semantic regions, stabilizing historical domain embeddings. The intra- and inter-domain disentanglement and historical embedding stability enable the ridge regression classifier to map disentangled and stable embeddings to labels via closed-form recursive updates, facilitating incremental updates while ensuring optimal knowledge preservation. Thus, distancing new domains from old prototypes prevents overlap in the stabilized embedding space, ensuring optimal knowledge retention by isolating domain semantics while facilitating incremental updates. Further explanation can be obtained in Theorem 3 of the response to reviewer #HUJf.
>
> ---
>
> > **Q3: Missing related references (Weakness 4)**
> >
>
> **A3:** Thank you for reminding this omission. We will add the suggested reference in the revised version.
>
> ---
>
> > **Q4: Performance in domains with little overlap and limitations in specific domains (Question 1)**
> >
>
> **A4:** We appreciate the reviewer's insightful comment.
>
> - For graph data from domains that have very little overlap in their feature semantics, please refer to Group 1 of Figure G.2 in Appendix G. Although very little overlap in raw features, other methods suffer from domain confusion due to embedding shifts caused by parameter updates across stages. In contrast, our method consistently maintains clear boundaries between domains in each stage.
> - Regarding limitations, GraphKeeper is agnostic to domain semantics and performs robustly across divergent domains. However, when fine-tuning pre-trained homogeneous GNNs on heterogeneous graph domains, structural disparities, which unrelated to feature semantics, causing a marginal performance drop. This is a universal challenge in heterogeneous and homogeneous graph learning, independent of our framework and beyond the scope of our discussion.
>
> ---
>
> > **Q5: More discussion of related work limitations (Question 2 & 4)**
> >
>
> **A5:** Thank you for highlighting this point.
>
> - GraphKeeper fundamentally differs from prior methods through its distinctive design for Domain-IL scenarios. Existing approaches (e.g., DeLoMe, PDGNNs) are confined to Task-IL or Class-IL settings, relying on end-to-end updates of a single GNN model. However, in Domain-IL, significant structural and semantic gaps across graph domains prevent a unified GNN from retaining multi-domain knowledge, rendering conventional designs ineffective. We attribute their failure to embedding shifts (distortion of prior domain embeddings) and decision boundary deviations (classifier instability).
>
> - GraphKeeper addresses this via: (1) Domain-specific PEFT atop a pre-trained GNN, isolating parameters to protect prior domains from interference; (2) The decision modules updated through ridge regression’s closed-form solution, rather than end-to-end backpropagation to update the entire model, preventing boundary drift while preserving knowledge; (3) Domain-aware distribution discrimination in high-dimensional space to assign PEFT parameters for inference on graphs with unobservable domain. Although TPP introduced a similar concept, its design remains confined to traditional scenarios, where the simple task-ID prediction mechanism and prompt design substantially diminish its effectiveness in the Domain-IL setting, which is validated by the performance comparisons in Table 1 of the paper.
>
> ---
>
> > **Q6: Discussion on the scalability of GraphKeeper (Question 3)**
> >
>
> **A6:** We appreciate the reviewer's insightful comment.
>
> **(1) Regarding the scalability to a large number of domains:**
> - The Table E.4 in Appendix demonstrate that GraphKeeper maintains superior performance even with 10+ incremental graph domains.
> - In terms of memory and computational costs, our experiments in Figure 7 of the paper validate that GraphKeeper achieves competitive performance even with low-rank PEFT modules, indicating that domain-specific parameters account for only a minor fraction compared to the pre-trained backbone. To quantify this, we tracked the total parameters and trainable parameters across incremental stages. Results show that adding a new domain increases total parameters by merely 5.2% relative to the pre-trained backbone, with only the specific PEFT parameters of current domain requiring training. This confirms that parameter growth slightly as the number of domains grows, and only a small subset of parameters is ever trained, ensuring memory and computational costs stay manageable even for large incremental domain sets.
>
> |  | Pretrained | 1 domain | 5 domains | 10 domains | 20 domains |
> | --- | --- | --- | --- | --- | --- |
> | Total Parameters | 196.61K | 206.85K | 247.81K | 299.01K | 401.41K |
> | Tuned Parameters |  | 10.24K | 10.24K | 10.24K | 10.24K |
>
> **(2) Regarding the sharing of PEFT modules:** this is an insightful comment that highlights the potential for leveraging similar knowledge from prior domains to facilitate forward transfer in incremental learning. As the first work to explore Graph Domain-IL, GraphKeeper prioritizes addressing catastrophic forgetting in this scenario, which is the most fundamental and critical capability for incremental learning. The cross-domain parameter sharing mechanism introduces new challenges, such as extracting domain-invariant knowledge and mitigating negative transfer. Therefore, while promising, this direction constitutes future work beyond the scope of this work.

---

> > ### Comment · Reviewer_4hfB · 2025-08-06
> >
> > Thank the authors for their detailed response, and most of my concerns have been solved. This paper offers an effective and well-motivated solution to domain-incremental learning, and its integration with GFMs provides much insights to the graph learning community.

---

> > > ### Author Response · Authors · 2025-08-08
> > >
> > > Thank you again for your valuable feedback and continued engagement throughout the discussion. As the rebuttal phase deadline is approaching, we would like to kindly check if you have any remaining concerns. If everything has been addressed, we would greatly appreciate it if you could help finalize the review by editing your original comments and filling in the Final Justification section.
> > >
> > > Thank you again for your constructive engagement!

---

> > > > ### Comment · Reviewer_4hfB · 2025-08-08
> > > >
> > > > The authors have addressed my concerns, and I will keep the positive score.

---

### Official Review · Reviewer_E6bP · 2025-07-03

**Clarity:** 2
**Significance:** 3
**Originality:** 3
**Rating:** 4
**Confidence:** 3

**Summary:**

Most existing work in graph incremental learning focuses on task- and class-increment settings within the same domain, while graph domain-incremental learning (DomainIL) remains challenging and underexplored. This paper focuses on the DomainIL scenarios, and proposes a framework, GraphKeeper, to address the catastrophic forgetting issue in DomainIL. The DomainIL framework includes a domain-specific parameter-efficient fine-tuning (PEFT) module to avoid embedding drift, a deviation-free knowledge preservation model to stabilize the decision boundary across domains, and a domain-aware distribution discrimination module to produce precise embeddings for unseen domains.

**Questions:**

- Are the reported results based on inference conducted on an unseen domain?
- In the integration studies with GFMs, why do you first pre-train the GFMs and how?
- Is it always beneficial or fair to push embeddings from different domains apart? Could you provide more discussion and explanation?
- How is the domain prototype selected? Is it computed as the average of all node embeddings within a domain?

**Ethical Concerns:**

["NO or VERY MINOR ethics concerns only"]

**Final Justification:**

The authors addressed my concerns in the rebuttal, and I have decided to keep my score at 4 (borderline accept) as it is already positive.

**Limitations:**

N/A.

**Paper Formatting Concerns:**

N/A.

**Quality:**

3

**Strengths And Weaknesses:**

Strengths:
- The paper is the first to explore the challenging problem of graph domain-incremental learning (DomainIL) and effectively motivates its importance.
- The proposed methods are technically sound. To mitigate catastrophic forgetting in DomainIL, the authors introduce domain-specific PEFT modules to prevent embedding shifts, and use Intra- and inter-domain disentanglement to make embeddings more discriminative across domains. On the other hand, the framework decouples the embedding model from the task classifier to preserve a stable decision boundary, which the classifier is updating via ridge regression and without historical data access.
- The proposed framework can be adapted to new, unseen domains, by the effect of a domain-aware distribution discrimination module.
- The experimental results are promising. In particular, it is interesting to observe that the proposed method brings significant improvement to GFMs in the DomainIL setting.

Weaknesses:
- The paper does not discuss Mixture-of-Experts (MoE) approaches for graph incremental learning. It would be helpful to compare with such methods, especially in relation to the proposed domain-specific graph PEFT design.
- The domains used during the incremental training process, their ordering, and the domain chosen for inference are not clearly specified.
- The explanation of how GraphKeeper integrates with GFMs lacks clarity. Specifically, it is unclear how GFMs are pre-trained, and how few-shot learning is conducted.
- It would be better to include the comparison between using GFMs with GraphKeeper and directly fine-tuning GFMs on new domains.
- Minor: Some references to figures or sections (e.g., Figure G.2, G.3) are broken or lead to incorrect locations.

---

> ### Author Rebuttal · Authors · 2025-07-28
>
> We appreciate your efforts and insightful comments. Below, we provide more detailed explanations to address your concerns.
>
> ---
>
> > **Q1: More discussions on related work (Weakness 1)**
> >
>
> **A1:** We are grateful for the reviewer's constructive suggestion. We have further investigated recent Mixture-of-Experts and parameter-efficient fine-tuning approaches relevant to graph incremental learning and incorporated comparisons into our analysis. Below, we clarify distinctions between GraphKeeper and existing methods:
> ﻿
>
> - DyMoE [1] memorizes knowledge of incremental classes through distinct experts, integrating the relevant knowledge via sparse MoE activation to enable positive transfer from previously learned knowledge. However, DyMoE operates exclusively within single domains, prioritizing task-level knowledge transfer, where its expert related integration fails under Domain-IL's core cross-domain structural and semantic gaps. GraphKeeper's domain-specific PEFT structurally resembles MoE but fundamentally differs, where we isolate domain parameters to prevent embedding shifts in incremental stages and achieve knowledge disentanglement between different domains, addressing Domain-IL's unique challenge of inter-domain interference.
> - FastKGE [2] leverages incremental LoRA (IncLoRA) to efficiently learn new knowledge graph embeddings while preserving old knowledge. However, its adapter allocation relies on entity-level influence within a single KG, overlooking the domain-level shifts critical to Domain-IL, which is a more challenging scenario requiring knowledge stability across heterogeneous domains. Although both adopt structure similar to incremental LoRA, GraphKeeper’s PEFT is fundamentally domain-centric, designed to stabilize and disentangle multi-domain knowledge. Meanwhile, its implementation is easily compatible (e.g., substituting LoRA with corresponding prompt strategy in GFMs), demonstrating flexibility and adaptability.
> - TPP [3] predicts task-ID via Laplacian smoothing, sharing conceptual similarity with our domain assignment. However, it faces significant risks in Domain-IL due to cross-domain semantic gaps ignored. In contrast, GraphKeeper’s high-dimensional distribution discrimination resolves prototype confusion, as shown in Figure 7 of the paper. What's more, TPP’s simplistic prompt design shows limited adaptability to multi-domain graphs, which evidenced by its average accuracy in Table 1 of the paper.
>
> ---
>
> > **Q2: More detailed explanation about experimental setup (Weakness 2, Question 1)**
> >
>
> **A2:** The incremental training domains, their ordering, and the inference settings are comprehensively specified in Appendix C.2.1 of the paper. To further clarify these aspects:
>
> - We constructed 8 incremental groups (Table C.2 in Appendix) from 15 diverse graph domains. Group 1-3 consist of graph domains from the same type (e.g., all are social networks in a group), while Group 4-6 include graph domains from mixed types. Groups 7–8 include over 10 graph domains to evaluate performance under longer incremental learning sequences.
> - To eliminate order bias, each group of domains was evaluated under all cyclic permutations of the task sequence. Such as for group of domains {A, B, C, D}, we evaluate in incremental orders of A → B → C → D, B → C → D → A, C → D → A → B, and D → A → B → C, where each order is independently executed 10 times. This design ensures robustness against sequence-dependent performance variance, with results averaged over all permutations.
> - For the evaluation protocol, we strictly adhere to incremental learning principles. During the process of incremental learning, an accuracy matrix ${M}$ is generated, where ${M}\_{i,j}$ represents the accuracy on $j$-th domain after learning $i$-th domain. Average Accuracy (AA) is computed as $\frac{1}{T}\sum\_{j=1}^{T} M\_{T,j}$, which represents the average performance across all domains after learning all domains. Average Forgetting (AF) is computed as $\frac{1}{T-1}\sum\_{j=1}^{T-1}(M\_{T,j} - M\_{j,j})$, which represents the difference between the final performance on each domain and the performance on the domain when it was first learned.
> - To ensure absolute fairness in incremental learning, the model is unaware of which domain the data to be evaluated belongs to during evaluation (i.e., graphs with unobservable domains in the paper). It should be emphasized that, the test graph belongs to one of the domains in the training process, rather than entirely new domains.
>
> ---
>
> > **Q3: More detailed explanation about the integration studies with GFMs (Weakness 3 & 4, Question 2)**
> >
>
> **A3:** We are grateful for the reviewer's concern to this detail.
>
> - GraphKeeper integrates with GFMs during the downstream fine-tuning phase, without altering their pretraining procedures, preserving their pretrained knowledge. GFMs are pretrained following the method proposed in corresponding paper, and pretraining datasets remain disjoint from downstream data, adhering to standard GFM settings.
> - Our experiments integrating GFMs focus on preserving their few-shot capability under low-resource constraints while endowing them with incremental learning and memory abilities, enabling continuous few-shot adaptation across multiple graph domains without forgetting. We replace the pre-trained GNN in our framework with pre-trained GFMs and adapt domain-specific PEFT modules. Concretely, for MDGPT, we replace the LoRA in PEFT module with its prompt-based strategy during fine-tuning. For GCOPE, we retain the original PEFT module and fine-tune the pre-trained GFM, consistent with its original design.
> - For the few-shot experimental setup, we sample 10 nodes per class per graph domain to effectively conduct incremental multi-domain downstream fine-tuning (as stated in Appendix C.2.1).
> - Table 2 in the paper already includes the performance of GFMs in continuous multi-domain downstream fine-tuning with/without the GraphKeeper framework. It shows that:
>     - Without GraphKeeper: The low AA and low AF of GFMs indicate that although GFMs achieve excellent performance in single-domain few-shot learning, they suffer from severe forgetting of prior domain knowledge during domain incremental few-shot learning.
>     - With GraphKeeper: GFMs perfectly integrate continual learning and memory abilities, while retaining few-shot capabilities, exhibiting high AA and near-zero AF.
>
> ---
>
> > **Q4: Broken reference locations (Weakness 5)**
> >
>
> **A4:** Thanks for the careful reminder. We will correct it in the revised version.
>
> ---
>
> > **Q5: More detailed explanation about disentanglement mechanism (Question 3)**
> >
>
> **A5:** We appreciate the reviewer's insightful comment. Pushing embeddings from different domains apart is synergistic with other components of GraphKeeper. First, intra-domain disentanglement in Eq. (5) and inter-domain disentanglement in Eq. (6) map nodes of distinct classes to unique positions in the semantic space, eliminating semantic confusion. Second, after learning each incremental domain, its domain-specific parameters are frozen, locking the embeddings within stable regions of the semantic space. This stability aligns perfectly with the deviation-free ridge regression classifier in Eq. (10) and Eq. (11), which maps different semantic regions to labels via closed-form updates, leveraging the disentangled and stable embeddings. Consequently, the synergy among these mechanisms ensures both zero forgetting of old domains and effective memory of new domain knowledge. Further explanation can be obtained in Theorem 3 of the response to reviewer #HUJf.
>
> ---
>
> > **Q6: More detailed explanation about the domain prototype (Question 4)**
> >
>
> **A6:** We are grateful for the reviewer's concern to this detail.
>
> - As described in Eq. (13) and Eq. (14), the domain prototype is computed as the average of node features within a high-dimensional space. Specifically, to mitigate assignment errors caused by potential overlaps among multiple domains, we first project node features into a more discriminative high-dimensional space via random mapping in Eq. (13), a concept similarly explored in literature [4-6]. The prototype for each domain is then obtained by averaging these high-dimensional features of all its training nodes. During evaluation, for a test graph with an unobservable domain, we similarly average the high-dimensional features of its test nodes to form a unified prototype, which is then matched against prototypes of all previously trained domains for precise domain-specific parameter assignment. The distance between prototypes is then used to estimate the probability of the test graph belonging to each domain. The more explanation can be obtained in Theorem 1-2 of the response to reviewer #kEra.
>
> - Furthermore, although using node embeddings for domain prototype calculation is also feasible due to the stability of node embeddings in our method, this requires performing inference on the test graph with all domain-specific parameters to compute multiple matching scores for assignment during testing, which incurs unnecessary computational overhead. Therefore, we adopt node features rather than the embeddings output by the model.
>
> ---
>
> > **References**
>
> [1] Kong L, et al. Dynamic Mixture-of-Experts for Incremental Graph Learning.
>
> [2] Liu J, et al. Fast and continual knowledge graph embedding via incremental lora. IJCAI 2024.
>
> [3] Niu C, et al. Replay-and-forget-free graph class-incremental learning: A task profiling and prompting approach. NeurIPS 2024.
>
> [4] McDonnell M D, et al. Ranpac: Random projections and pre-trained models for continual learning. NeurIPS 2023.
>
> [5] Sun J, et al. Feature expansion for graph neural networks. ICML 2023.
>
> [6] Zhuang H, et al. Acil: Analytic class-incremental learning with absolute memorization and privacy protection. NeurIPS 2022.

---

> ### Comment · Area_Chair_K6DS · 2025-08-07
>
> Please leave your feedback for the authors' rebuttal to avoid being flagged as an irresponsible reviewer.

---

> ### Comment · Reviewer_E6bP · 2025-08-07
>
> Thank you to the authors for the response. It addressed most of my concerns. I will keep my previous scores.

---

> > ### Author Response · Authors · 2025-08-08
> >
> > Thank you again for your valuable feedback and continued engagement throughout the discussion. As the rebuttal phase deadline is approaching, we would like to kindly check if you have any remaining concerns. If everything has been addressed, we would greatly appreciate it if you could help finalize the review by editing your original comments and filling in the Final Justification section.
> > ﻿
> > Thank you again for your constructive engagement!

---

### Official Review · Reviewer_HUJf · 2025-07-07

**Clarity:** 3
**Significance:** 2
**Originality:** 3
**Rating:** 4
**Confidence:** 4

**Summary:**

This paper addresses the problem of Graph Domain-Incremental Learning (Domain-IL), where a graph model is required to learn sequentially from graphs belonging to different domains without forgetting previous knowledge. The authors identify two key challenges, embedding shifts and decision boundary drift, that lead to catastrophic forgetting in this setting. To address these, they propose GraphKeeper, a framework that combines domain-specific parameter-efficient fine-tuning, intra- and inter-domain disentanglement, and a ridge regression-based classifier to stabilize knowledge across domains. Experiments on 15 graph datasets show that the proposed method achieves consistent improvements over existing graph incremental learning baselines.

**Questions:**

- The proposed method focuses on preventing forgetting through domain-specific parameter isolation but does not address how knowledge can be shared or transferred across related domains. Have the authors considered mechanisms to enable forward transfer or knowledge accumulation in Domain-IL settings?

- Can the authors provide further analysis or experiments to assess the robustness of this domain assignment and its impact on downstream performance?

**Ethical Concerns:**

["NO or VERY MINOR ethics concerns only"]

**Final Justification:**

I decided to raise my score as the authors have provided thorough theoretical and empirical justifications addressing my concerns, particularly regarding knowledge transfer, robustness under domain similarity, and scalability. Their clarifications support the paper's contributions to domain-incremental learning on graphs.

**Limitations:**

The key limitations are summarized in the weaknesses section above.

**Paper Formatting Concerns:**

No concerns

**Quality:**

2

**Strengths And Weaknesses:**

Strengths

- The paper is the first to systematically address Graph Domain-Incremental Learning (Domain-IL), a practically relevant yet underexplored setting in graph learning.

- The proposed combination of domain-specific parameter-efficient tuning (PEFT) and ridge regression-based knowledge preservation is well-motivated, computationally efficient, and compatible with graph foundation models.

- Extensive experiments on 15 diverse graph datasets, combined with careful ablations and visualization, consistently demonstrate the effectiveness of the method in mitigating catastrophic forgetting.

Weaknesses

- The proposed framework isolates parameters for each domain to prevent forgetting, but this design inherently limits positive transfer across related domains. There is no mechanism to extract, share, or accumulate transferable knowledge when structural or semantic similarities exist between domains. While the method effectively mitigates forgetting, it overlooks forward transfer and knowledge accumulation, which are essential components of continual learning.

- The method assumes access to domain labels during training and uses heuristic prototype matching for domain assignment at test time. This approach can be fragile when domains are similar, potentially leading to incorrect parameter selection and degraded performance. The paper does not analyze or address the impact of such domain misclassification.

- While the paper provides formal derivations for its classifier update and disentanglement objectives, it lacks a deeper theoretical justification or generalization analysis explaining why these mechanisms prevent forgetting in the Domain-IL setting.

- Maintaining separate PEFT modules for each domain may incur significant memory and computational costs as the number of domains grows, which is not discussed.

---

> ### Author Rebuttal · Authors · 2025-07-28
>
> We appreciate your comments. To address your concerns, below we provide further analysis and experiments.
>
> ---
>
> > **Q1: The knowledge transfer across related domains (Weakness 1, Question 1)**
> >
>
> **A1:** We sincerely appreciate your insightful comment on cross-domain knowledge transfer in domain-IL settings.
>
> - We fully agree that forward transfer is promising for incremental learning. But importantly, Graph Domain-IL has no prior work, where the fundamental challenge is catastrophic forgetting caused by structural and semantic gaps, which is the most essential capability for incremental learning. As the first exploration in this underexplored setting, GraphKeeper establishes a foundational anti-forgetting framework, which is the main focus and contribution of our work.
>
> - Regarding cross-domain knowledge transfer, we observed that our method exhibited positive transfer in the incremental learning process of some graph domains (AF was positive, as shown in Table 1 of the paper). It should be emphasized that, despite domain-specific parameter isolation, nodes from all incremental domains are embedded into a unified semantic space. Consequently, as our closed-form ridge regression classifier employs recursive updates in the new domain, it triggers adaptive adjustments to the decision boundaries of previous graph domains. This collaborative boundary evolution demonstrates the potential for implicit positive transfer. We fully agree with your vision regarding positive transfer and are actively exploring mechanisms for cross-domain parameter sharing while combining them with domain-specific parameters. Since this requires addressing new challenges in domain-invariant knowledge and negative transfer, it constitutes future work to be pursued and falls outside the scope of this work.
>
> ---
>
> > **Q2: The impact of similar domains for Domain-aware Distribution Discrimination (Weakness 2, Question 2)**
> >
>
> **A2:** We thank the reviewer for raising this insightful concern. Below, we provide a systematic response through theoretical theorems and supplementary experiments. The supplementary analyses will be incorporated into the revised version.
>
> **Theorem 1. Linear Separability via Random High-Dimensional Mapping:** Let $D_i$ and $D_j$ be feature distributions of two graph domains with overlap $\kappa \in [0,1]$. For a fixed random mapping $\phi:\mathbb{R}^{d} \rightarrow \mathbb{R}^{m}$, where $m \gg d$, the mapped features satisfy:
>
> $$
> \mathbb{P}\left[\left\| \phi(\boldsymbol{x}_i \sim \mathcal{D}_i) - \phi(\boldsymbol{x}_j \sim \mathcal{D}_j) \right\|_2 \geq \Delta \right] \geq 1 - \delta
> $$
>
> where $\Delta >0$ is a separation margin, and $\delta$ decays exponentially with $m$. Even when $D_i$ and $D_j$ overlap to a large extent, random projection amplifies distributional differences in high-dimensional space, ensuring linear separability with high probability if $\kappa \neq 1$. This stems from the Johnson-Lindenstrauss Lemma and geometric properties of random projections.
>
> **Theorem 2. Stability of Prototype Matching:** Let $\boldsymbol{x}\_{\text{test}} \sim \mathcal{D}\_{k}$ be the test node features from domain $k$, and $\boldsymbol{p}\_{k} = \mathbb{E}\_{\boldsymbol{x}\_{\text{train}} \sim \mathcal{D}\_{k}}[\phi(\boldsymbol{x}\_{\text{train}})]$ be the true domain prototype. Given a fixed random mapping $\phi$ and identical train and test distributions:
>
> $$
> \left\| \boldsymbol{p}_{\text{test}} - \boldsymbol{p}_k \right\|_2 \leq \epsilon_1
> $$
>
> For mismatched domain prototypes $\boldsymbol{p}_{j}$, where $j \neq k$:
>
> $$
> \left\| \boldsymbol{p}_{\text{test}} - \boldsymbol{p}_j \right\|_2 \geq \Delta - \epsilon_2
> $$
>
> where $\Delta$ is the separation margin from Theorem 1. Since the test graph's nodes share the same distribution as the training nodes of its source domain, test graphs converge to their true domain prototype $\boldsymbol{p}\_k$ (i.e. $\epsilon_1$ tends towards zero). Meanwhile, even when the feature distributions across domains exhibit substantial overlap but are not identical (i.e. $\kappa \neq 1$), coupling this with a sufficiently high mapping dimension ensures that $\Delta - \epsilon_2$ tends to be sufficiently large. This guarantees that the lower bound of $\left\| \boldsymbol{p}\_{\text{test}} - \boldsymbol{p}\_j \right\|\_2$ is always greater than the upper bound of $\left\| \boldsymbol{p}\_{\text{test}} - \boldsymbol{p}\_k \right\|\_2$.
>
> **Supplementary experiments.** We conducted robustness evaluations under different domains with high distributional overlap. Specifically, we randomly sampled graphs from four mutually similar distributions as four graph domains, where the pairwise domains feature distribution overlap exceeded 80%. For each domain, we partitioned training and test nodes to derive domain prototypes and test graph prototypes. The confusion matrix below shows the probability of our method correctly predicting the source domain of each test graph.
>
> ||$\boldsymbol{p}_{1}$|$\boldsymbol{p}_{2}$|$\boldsymbol{p}_{3}$|$\boldsymbol{p}_{4}$|
> |---|---|---|---|---|
> |$\boldsymbol{p}_{1}^{\text{test}}$|**38.16%**|22.77%|18.91%|20.14%|
> |$\boldsymbol{p}_{2}^{\text{test}}$|21.02%|**38.94%**|21.30%|18.72%|
> |$\boldsymbol{p}_{3}^{\text{test}}$|19.57%|23.72%|**39.79%**|16.91%|
> |$\boldsymbol{p}_{4}^{\text{test}}$|23.49%|20.47%|16.49%|**39.52%**|
>
> These results demonstrate that even when confronted with highly similar cross-domain distributions, our method ensures that the matching probability between test graph prototypes and their source domain prototypes is significantly higher than that with other domains, thereby enabling accurate and robust domain-specific parameter assignment.
>
> ---
>
> > **Q3: The analysis for GraphKeeper to prevent forgetting in the Domain-IL setting (Weakness 3)**
> >
>
> **A3:** The stability of GraphKeeper in mitigating catastrophic forgetting stems from the synergistic operation of its core mechanisms.
>
> **Theorem 3. Stability Guarantee of GraphKeeper:** Let $\mathcal{Z}$ denote the disentangled semantic space. For any two domains $\mathcal{D}\_u, \mathcal{D}\_v$ and classes $\mathcal{C}\_i^u,\mathcal{C}\_j^u \in \mathcal{D}\_u$, the disentangled embeddings $\mathbf{z} \in \mathcal{Z}$ satisfy:
>
> $$
> \min\_{\substack{\mathbf{z}\_1 \in \mathcal{C}\_i^u, \mathbf{z}\_2 \in \mathcal{C}\_j^u}} \|\mathbf{z}\_1 - \mathbf{z}\_2\|_2 \geq \tau\_{\text{intra}}, \quad \min\_{\substack{\mathbf{z}\_1 \in \mathcal{C}\_i^u, \\ \mathbf{z}\_2 \in \mathcal{C}\_j^v}} \|\mathbf{z}\_1 - \mathbf{z}\_2\|_2 \geq \tau\_{\text{inter}}
> $$
>
> where intra- and inter-domain disentanglement objectives in Eq. (5) and Eq. (6) ensure that $\tau\_{\text{intra}}$ and $\tau\_{\text{inter}}$ are sufficiently large, which collocates nodes of distinct classes into unique semantic location, forming compact intra-class clusters with clear boundaries. The domain-specific PEFT then freezes these semantic regions, stabilizing historical domain embeddings.
>
> The ridge regression classifier $\psi$ establishes a stable mapping from the disentanglement semantic space $\mathcal{Z}$ to the label space $Y$:
>
> $$
> \psi\_{t}(\mathbf{z},\boldsymbol{W}\_{t}):\mathcal{Z}\rightarrow Y, \quad \boldsymbol{W}\_{t} = \arg\min\_{\boldsymbol{W}}\left\|\boldsymbol{Y}\_{(1:t)}-\mathcal{Z}\_{(1:t)}\boldsymbol{W}\right\|\_{\mathrm{F}}^2+\lambda\left\|\boldsymbol{W}\right\|_{\mathrm{F}}^2,
> $$
>
> where the parameters $\boldsymbol{W}$ of the ridge regression classifier $\psi$ satisfy the optimal solution at any incremental stage, which is guaranteed by the derivation of the ridge classifier’s exact historical knowledge retention in Appendix A. The intra- and inter-domain disentanglement and historical embedding stability enable the ridge regression classifier to map disentangled and stable embeddings to labels via closed-form recursive updates.
>
> For inference, given test data $\boldsymbol{x}\_{\text{test}} \sim \mathcal{D}\_k$, the training data  $\boldsymbol{x}\_{\text{train}}$ of source domain is also identically distributed. Due to Theorem 1-2 guarantee domain-specific parameter assignment is accurate and robust, the prediction error:
>
> $$
> \lim_{\tau\_{\text{intra}},\tau\_{\text{inter}} \to +\infty} \mathbb{P}\left( \psi(\mathbf{z}\_{\text{train}}) \neq y\_{\text{train}} \right) = 0, \quad  \lim_{\epsilon \to 1} \mathbb{P}\left( \psi(\mathbf{z}\_{\text{test}}) \neq y\_{\text{test}} \right) = \mathbb{P}\left( \psi(\mathbf{z}\_{\text{train}}) \neq y\_{\text{train}} \right)
> $$
>
> where the prediction error of training data converges to zero when $\tau\_{\text{intra}}$ and $\tau\_{\text{inter}}$ are sufficiently large. Meanwhile, the prediction error of the test data converges to the same level as that of the training data of source domain when the distribution matching degree $\epsilon$ in the embedding space $\mathcal{Z}$ approaches 1.
>
> ---
>
> > **Q4: The analysis of memory and computational costs as the number of domains grows (Weakness 4)**
> >
>
> **A4:** We are grateful for the reviewer's concern to this detail.
>
> Our experiments in Figure 7 of the paper validate that GraphKeeper achieves competitive performance even with low-rank PEFT modules, indicating that domain-specific parameters account for only a minor fraction compared to the pre-trained backbone. To quantify this, we tracked the total parameters and trainable parameters across incremental stages.
> ||Pretrained|1 domain|5 domains|10 domains|20 domains|
> |---| --- | --- | --- | --- | --- |
> |Total Parameters|196.61K|206.85K|247.81K|299.01K|401.41K|
> |Tuned Parameters||10.24K|10.24K|10.24K|10.24K|
>
> Results show that adding a new domain increases total parameters by merely 5.2% relative to the pre-trained backbone, with only the specific PEFT parameters of current domain requiring training. This confirms that parameter growth slightly as the number of domains grows, and only a small subset of parameters is ever trained, ensuring memory and computational costs stay manageable even for large incremental domain sets.

---

> ### Comment · Area_Chair_K6DS · 2025-08-07
>
> Please leave your feedback for the authors' rebuttal to avoid being flagged as an irresponsible reviewer.

---

### Note · Authors · 2025-08-15

We sincerely appreciate all the reviewers and ACs﻿ for the time and effort invested in reviewing process! We appreciate the reviewers' recognition of our contributions, which can be summarized as follows:
﻿
﻿
- **The First Graph Domain Incremental Learning Framework:** We are the first to explore this challenging and important scenario, and we analyze the unique challenges of catastrophic forgetting within it. [`#HUJf, #E6bP, #4hfB, #kEra, #CzLz`]
﻿
﻿
- **Novel perspective and method design:** We address catastrophic forgetting in the Domain-IL scenario through multi-domain graph disentanglement and deviation-free knowledge preservation, offering a novel perspective. [`#HUJf, #E6bP, #4hfB, #kEra`]
﻿
﻿
- **Comprehensive Experiments and Superior Performance:** We conducted extensive experiments on diverse datasets, including more challenging settings like extremely long incremental sequences. These demonstrate our method's state-of-the-art performance alongside the scalability and stability. [`#HUJf, #E6bP, #4hfB, #CzLz`]
﻿
﻿
- **Compatibility with Graph Foundation Models:** Our method integrates effectively with existing GFMs, thereby equipping them with the ability to continually integrate knowledge while preserving their inherent few-shot adaptation capabilities, to construct more powerful GFMs. [`#HUJf, #E6bP, #4hfB, #CzLz`]


Although some concerns were raised, **we have addressed all concerns thorough detailed point-by-point responses during the rebuttal phase, and all reviewers acknowledged our clarifications.** Ultimately, **all reviewers assigned positive ratings, representing a unanimous intention for the paper to be accepted.**
﻿

We once again express our gratitude to all the reviewers and ACs﻿﻿ for their time and effort!

---

### Decision · Program_Chairs · 2025-09-17

**Decision:**

Accept (poster)

**Comment:**

This paper introduces GraphKeeper, a framework for Graph Domain-Incremental Learning (Domain-IL), where models must continuously learn from graphs across different domains without forgetting prior knowledge. The authors identify embedding shifts and decision boundary drift as the main causes of catastrophic forgetting and address them through domain-specific parameter-efficient fine-tuning, intra- and inter-domain disentanglement, and ridge regression–based knowledge preservation. Experiments on 15 datasets demonstrate that GraphKeeper consistently outperforms existing incremental learning baselines, offering stable embeddings, robust decision boundaries, and improved accuracy in multi-domain graph learning.

Strengths:
1. Most reviewers find the paper well-motivated and recognize its novelty in addressing graph domain-incremental learning.
2. Most reviewers consider the experiments extensive and the results promising.

Weakness:
1. Some reviewers raised concerns about the transferability of the proposed method across similar domains
2. Some reviewers ask for theoretical justification.

During the rebuttal, the authors have provided:
1. New theoretical justification
2. Additional experiments to justify model's scalability, knowledge transferability and robustness under domain similarity.

The authors addressed all concerns during the rebuttal, and all reviewers responded with positive ratings, indicating unanimous support for acceptance.